# Robust antibiotic sensitization of pathogenic *Pseudomonas aeruginosa* via negative hysteresis in the cell envelope

Florian Buchholz[1,17], Lina M. Upterworth[1,17], Leif Tueffers[1,2], Espen E. Groth[3,4,5], Kira Haas[1], Daniel Schütz[1], Abigail Savietto Scholz[6], Aditi Batra[1], Surajit Pal[1], Samarpita Banerjee[1], Badri N. Dubey[7,8], Sören Franzenburg[9], Barbara Kalsdorf[10,11,12], Klaus F. Rabe[3,4,5], Dennis Nurjadi[2], Jan Rupp[2,11,13], Dan I. Andersson[14], Holger Sondermann[7], Marc Bramkamp[6], Roderich Roemhild[1,14,15,16,18] ✉ & Hinrich Schulenburg[1,16,18] ✉

Antibiotic combination in time and space is a key strategy to combat antimicrobial resistance. The success of such treatment designs requires their robust efficacy across treatment conditions and a pathogen's genomic diversity. This study found that an initial treatment with a β-lactam antibiotic causes robust cellular sensitization towards an aminoglycoside antibiotic across the high-risk human pathogen *Pseudomonas aeruginosa*, including resistant strains. This phenomenon of cellular sensitization, termed negative hysteresis, is modulated by the Cpx envelope stress response system and linked to membrane stress during growth. The increase in efficacy is achieved through a β-lactam induced elevated cellular uptake of the subsequently administered aminoglycoside. Negative hysteresis and the Cpx system are linked in several cases to the expression of synergistic drug interactions, thus enhancing efficacy of antibiotic combinations. Overall, our study identifies the phenomenon of negative hysteresis as a robustly inducible phenotype and thus a unique focus for optimizing antimicrobial therapy.

Antimicrobial resistance (AMR) has become a major threat to human health[1–3]. It was associated with 4.95 million deaths and directly accounted for 1.27 million deaths in 2019, while future projections suggest a further spread of AMR and an increase in mortality especially among the elderly[2,3]. *Pseudomonas aeruginosa* is a particularly problematic opportunistic human pathogen in this context[2,3]. It causes acute and chronic infections in hospitalized and immunocompromised patients and is a common pathogenic resident of the respiratory tract in patients with cystic fibrosis (CF) or chronic obstructive pulmonary disease (COPD). Due to the bacterium's high intrinsic resistance and the ability to rapidly adapt to new conditions (e.g., newly administered drugs), treating *P. aeruginosa* infections has become

challenging[1,4–6]. The World Health Organization (WHO) has thus listed *P. aeruginosa* as a high priority pathogen[7], for which new treatment options are urgently required. In addition to the costly and time-consuming development of new antimicrobials, multi-drug designs show particular promise as a sustainable counter to AMR, using either simultaneous or sequential application of antimicrobial drugs[1,8–11].

Sequential therapy has previously been shown to be effective against *P. aeruginosa* through independent in vitro studies[12–18]. One reason for its efficacy is negative cellular hysteresis, where exposure to one antibiotic sensitizes bacterial cells to the subsequently administered antibiotic upon drug switches[12]. Hysteresis describes the interactions of antibiotics across time and treatment sequence. In

contrast to the interaction profile in a combination treatment where two antibiotics are simultaneously present and may co-induce sensitivity mechanisms, sensitization in a sequential treatment may depend on the specific order of the antibiotics. In particular, one direction of an antibiotic switch may lead to sensitization (i.e., negative hysteresis) while the opposite direction could result in a protective effect (i.e., positive hysteresis)[19]. The possible directionality of drug interactions across antibiotic switches was first described in *E. coli* in 1962[20], but has since been lost from focus. Only recently has the effect gained new attention, with the discovery of directional and bidirectional negative hysteresis among common antibiotics in *P. aeruginosa*[12,13]. Generally, many synergies are driven by just one antibiotic of the combination pair, facilitating the activity of the other[21]. Therefore, directional, temporal hysteresis interactions, while understudied, are generally expected, and their presence is at least indicated in several pathogens that had been subjected to some form of antibiotic switching[22–24]. Negative hysteresis and antibiotic synergy are both caused by physiological stress of bacterial cells. They are thus clearly distinct from collateral sensitivity, where increased sensitivity towards a particular antibiotic has an acquired genetic basis and results from the earlier evolution of resistance towards another antibiotic by mutation or gain of a mobile element[12,19]. In addition, although hysteresis and drug interaction type are related, they are also distinct from one another, because hysteresis effects decay with time post induction when the inducing antibiotic is not present throughout. Despite this transience, negative hysteresis was found to increase bacterial extinction, constrain bacterial adaptation rates as well as reduce resistance evolution during a laboratory-based evolution experiment[12]. However, the previous study was focused on a single strain of *P. aeruginosa* (i.e., the laboratory strain PA14), few drugs (carbenicillin, gentamicin, ciprofloxacin), a narrow concentration space (a 75% inhibitory concentration, IC75), and it implicated a single mutation in the *P. aeruginosa* gene *cpxS*, which is part of the Cpx cell envelope stress response system, in contributing to hysteresis[12]. Therefore, to date, critical information for the general understanding of this phenomenon as well as its utilization for antibiotic therapy is missing, in particular regarding (i) which antibiotics and switching directions allow for optimal sensitization, (ii) how robust is negative hysteresis across the strain diversity known for *P. aeruginosa*, (iii) is antibiotic sensitization possible against resistant strains and diverse pathogen populations, and (iv) which role does the Cpx envelope stress response system play in sensitization?

Our study's overall objective is to establish a fundamental understanding of antibiotic sensitization via negative hysteresis. Thus, to address the current knowledge gaps, we (i) systematically characterized the conditions needed for antibiotic-induced cellular sensitization, (ii) inferred the presence, directionality, and robustness of hysteresis across antibiotics (including distinct β-lactam, fluoroquinolone, and aminoglycoside drugs), and also across the genomic diversity of *P. aeruginosa*, using the representative major *P. aeruginosa* clone type (mPact) panel[25–27], single-drug resistant populations[28], and diverse, newly isolated patient populations, and (iii) dissected the underlying mechanism by combining transcriptomics, a functional genetic analysis, and physiological characterizations of the Cpx envelope stress response system. Our results highlight the robust induction of negative hysteresis across *P. aeruginosa*, especially upon switches from β-lactam drugs to aminoglycosides, and additionally demonstrate a central involvement of the Cpx system in mediating cellular sensitization.

## Results

### Negative hysteresis is robustly induced in the laboratory strain PA14 upon switches from carbenicillin to gentamicin

We performed a thorough characterization of antibiotic sensitization in PA14, using the β-lactam antibiotics carbenicillin (CAR) and the aminoglycoside gentamicin (GEN). Hysteresis time-kill experiments were performed with varying antibiotic concentrations, switch orders, pre-treatment durations, and durations of the gap between pre- and main treatment (Fig. 1A). To characterize the dose dependency of hysteresis, we initially conducted experiments with a single well-growing culture across a wider range of concentrations (Supplementary Fig. S1A–C) and then validated our observations for selected conditions using repeat experiments (Fig. 1B–D). We found that concentrations below the minimal inhibitory concentration (i.e., sub-MIC concentrations) of the sensitizing drug CAR were sufficient to induce negative hysteresis and that the effect increased gradually with increasing CAR concentration (Fig. 1B, Supplementary Fig. S1B). Doubling the GEN concentration led to a saturation of bacterial killing independent of the pre-treatment concentration (Supplementary Fig. S1B). The increased killing of CAR pre-treated bacteria was not fully explained by the bacteria having spent more time under antibiotic treatment, as pre-treatment with GEN did not increase killing under GEN main treatment to the same degree as CAR pre-treatment (Fig. 1C, Supplementary Fig. S1A). When the order of drugs was reversed (i.e. pre-treatment with GEN, main treatment with CAR), killing was only observed for above MIC pre-treatment concentrations (Fig. 1D, Supplementary Fig. S1C). These findings confirmed the previously described sensitization of CAR pre-treated cells towards GEN and the directionality of the hysteresis effect[12]. Negative hysteresis was distinct from the post-antibiotic effect[29] as cells that received CAR pre-treatment behaved similarly to the no-drug control once the antibiotic was removed (Supplementary Fig. S2A, statistical results in Table S1). Removal of possible residual pre-treatment antibiotic by two consecutive washes with phosphate-buffered saline did not abolish a significant hysteresis effect (Supplementary Fig. S2B, statistical results in Table S2), consistent with negligible carry-over of unbound CAR into the main treatment. Covalently bound CAR and/or any CAR-induced changes in cellular physiology are thus likely to be sufficient for hysteresis. In addition, negative CAR-GEN hysteresis required actively growing cells since there was no hysteresis effect for bacteria that were growth arrested by strong chloramphenicol treatment that blocks new protein synthesis (Fig. 1E). At a pre-treatment concentration of 400 mg/L (4 x MIC), CAR exposure for 1 min was sufficient to induce negative hysteresis (Fig. 1F). Increasing the pre-treatment duration increased the hysteresis effect in the main treatment, consistent with the accumulation of CAR-induced cellular damage.

To address how long the CAR priming effect lasts, we varied the time during which the bacteria were allowed to grow in antibiotic-free medium between the pre- and main treatment (Fig. 1G). After approximately one generation of growth ($\approx 45$ min), the hysteresis effect started to vanish, and it more or less disappeared after a pause of 80 min (nearly 2 generations of growth). A possible explanation for this result might be a physiological reset of the CAR-stressed cell envelope after cell division.

### Negative hysteresis is induced by pre-treatment with diverse β-lactam drugs and further contributes to the synergism of antibiotic combinations

Next, we assessed which kind of antibiotic switches lead to negative hysteresis, using all possible bidirectional combinations of several clinically used antibiotics, including three β-lactams from different sub-classes (i.e., the carbapenem meropenem (MER); the penicillin antibiotic piperacillin combined with the β-lactamase inhibitor tazobactam (PIT); the cephalosporin ceftazidime (CTZ), the fluoroquinolone ciprofloxacin (CIP), and the aminoglycoside GEN. The results with standardized dosage are summarized in Fig. 2A as a heatmap, showing the mean difference in area under the curve (ΔAUC) of the time-kill curves between the hysteresis treatment (i.e., with a pre-treatment) versus the control treatment, and for all combinations, including a total of 120 time-kill curves. We found that β-lactam pre-

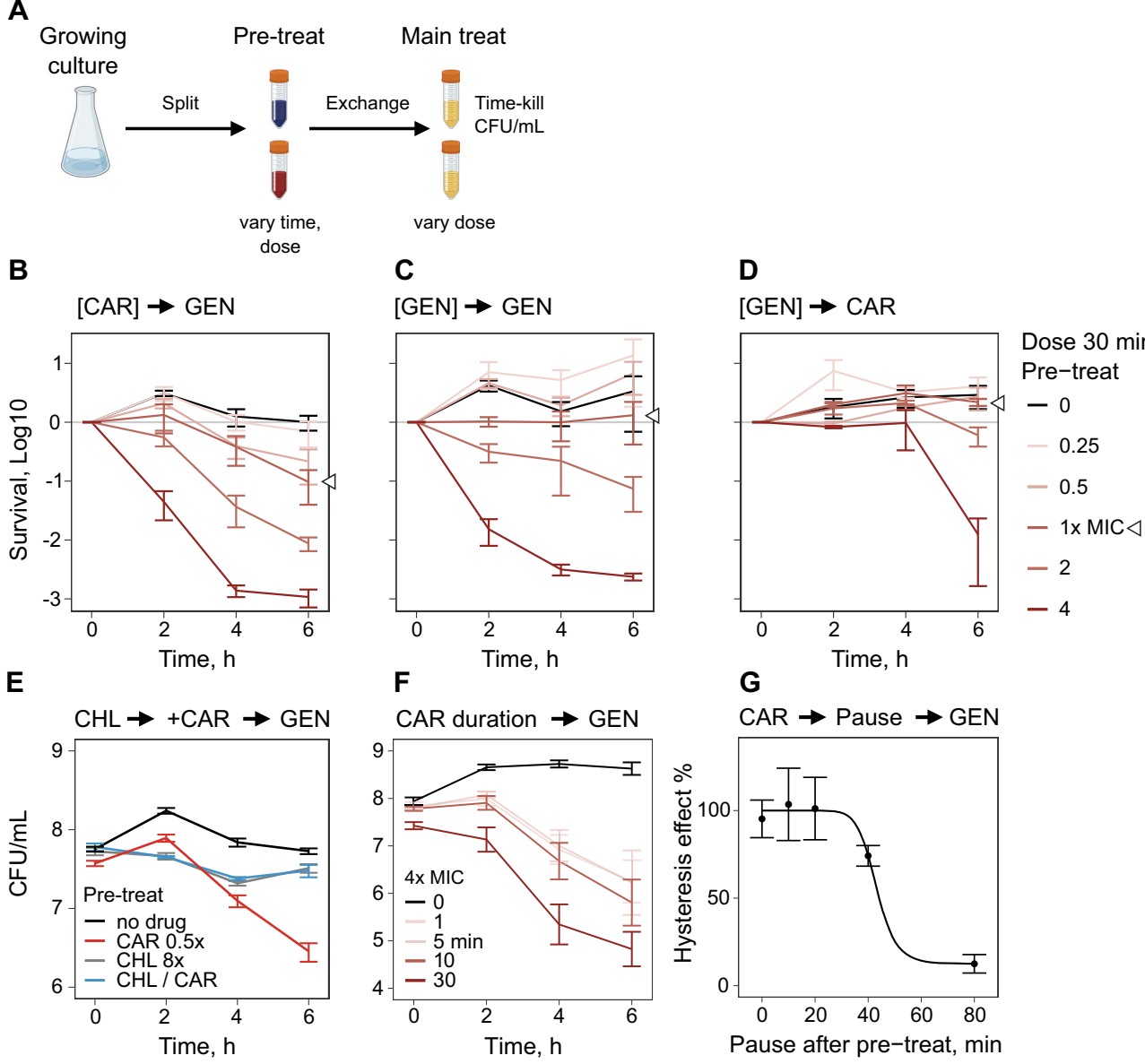

**Fig. 1 | Characterization of the hysteresis phenomenon in *Pseudomonas aeruginosa* PA14. A** Exemplary design of a hysteresis experiment. An exponential phase culture was split into sub-samples that are treated with different concentrations of the pre-treatment drug. After the pre-treatment the medium was exchanged and the main treatment drug was added followed by a time-kill experiment (Flask and tube icons: Created in *BioRender*. Upterworth, L. (2026) https://BioRender.com/9p03ozg). **B** Time-kill curves of 30 min carbenicillin (CAR) pre-treatment at various concentrations followed by gentamicin (GEN) main treatment ($n = 3$, mean ± standard error of the mean (SEM)). The hysteresis effect increased with increasing concentration of the pre-treatment antibiotic. **C** Time-kill curves of 30 min GEN pre-treatment followed by GEN main treatment ($n = 3$, mean ± SEM). Pre-treatment with the same drug increased killing during main treatment, although not to similar extent as for CAR pre-treatment (**B**). **D** Time-kill curves of 30 min GEN pre-treatment followed by CAR main treatment ($n = 3$,

mean ± SEM). Only high concentrations of GEN pre-treatment increased killing rates during CAR main treatment. **E** Negative hysteresis is only observed for growing cells. The sensitizing effect of CAR pre-treatment is suppressed when protein biosynthesis is blocked with the bacteriostatic antibiotic chloramphenicol (CHL) before the pre-treatment. (mean ± SEM, $n = 3$) The grey treatment (CHL 8x) is hidden under the blue treatment (CHL / CAR). **F** Time-kill curve of various durations of CAR pre-treatment followed by GEN main treatment at 1 mg/L (mean ± SEM; $n = 3$). Short pre-treatment was sufficient to induce the hysteresis effect. The effect size increased with pre-treatment duration. Long duration of pre-treatment at above MIC concentrations had an impact on the CFU/mL at the beginning of the main treatment. **G** Nelder-Mead dose-response model of a CAR-GEN time-kill experiment with pauses of different durations between the pre- and main treatment (mean ± SEM, $n = 4$). The priming effect of the pre-treatment disappeared over time. The data for this figure is provided in the supplementary Source data file.

treatment robustly induced negative hysteresis when followed by GEN (Fig. 2A). As with CAR, the observed strong negative hysteresis occurred across a wide range of subinhibitory and inhibitory β-lactam pre-treatment concentrations in a dose-dependent manner (Supplementary Fig. S1D–I with extended dosage analysis). Consistent with previous work[12], the reverse switch from GEN to the β-lactams resulted in positive hysteresis (Supplementary Fig. S1D–I). Switches between

CIP and GEN produced mild positive hysteresis when CIP was first and strong positive hysteresis when CIP was second. There was weak negative hysteresis for the switches from β-lactams to CIP, and mild varied responses in the reverse direction. Moreover, a switch between two β-lactams produced one case of negative hysteresis (i.e., PIT followed by CTZ) and otherwise positive hysteresis to varying degrees. These results highlight that even switches between two similarly acting

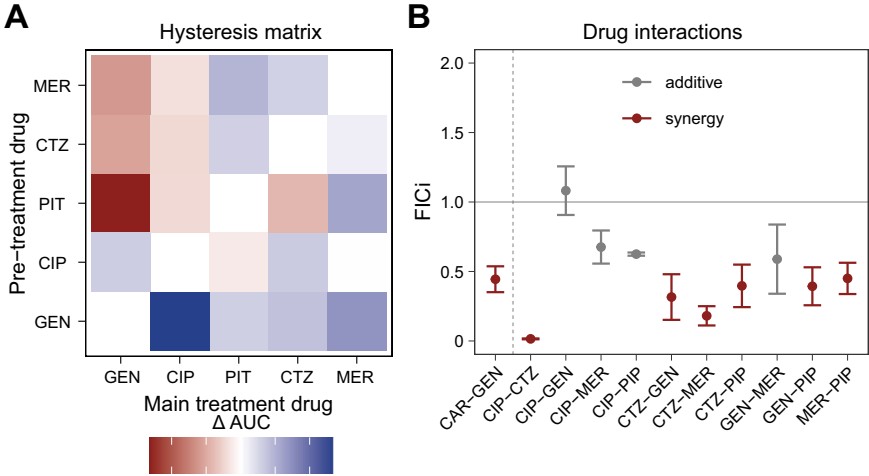

**Fig. 2 | Expression of hysteresis upon switches between clinically relevant antibiotics and its relationship to drug interaction phenotypes in *P. aeruginosa* PA14. A** Heatmap summarizing time-kill experiments assessing pair-wise hysteresis effects, inferred from differences between area under the curve (ΔAUC) of time-kill curves of hysteresis treatments (with respective pre-treatments) and controls (main-treatment as pre-treatment), using clinically relevant antibiotics for *P. aeruginosa*. β-lactam (MER, CTZ, PIT) pre-treatment followed by GEN main treatment consistently induced negative hysteresis. Changing the drug order resulted in positive hysteresis. There was weak negative hysteresis for β-lactams

followed by CIP. ($n = 3$). **B** Pair-wise drug interactions, shown as fractional inhibitory concentration index (FICi; synergy: FICi < 0.5; antagonism: FICi > 4; additive inter-action: 0.5 < FICi < 4), of the drug combinations used in (**A**). Antibiotic pairs with strong negative hysteresis effects were often but not always synergistic, while drug pairs with weaker hysteresis effects tended to be additive. (mean ± SD, $n = 3$–8). MER Meropenem, CTZ Ceftazidime, PIT Piperacillin/Tazobactam, CIP Cipro-floxacin, GEN Gentamicin, CAR Carbenicillin, PIP Piperacillin. The data for this fig-ure is provided in the supplementary Source data file.

drugs may cause negative hysteresis, as previously seen for other β-lactam drugs in *P. aeruginosa*[13].

In contrast to laboratory conditions where separation of pre- and main treatment is possible, medical implementation of a hysteresis approach would always have a phase where both drugs overlap due to pharmacokinetic processes, possibly resulting in drug interactions. Therefore, we assessed a possible effect of hysteresis on drug inter-actions and found that some drug pairs with strong negative hysteresis clearly enhanced each other's effect while applied simultaneously (i.e., synergistic interaction, for example for GEN-PIP or GEN-CTZ; Fig. 2B), while some with strong positive hysteresis showed an additive inter-action (especially GEN-CIP; Fig. 2B). In spite of these examples, other tested drug pairs do not indicate a direct translation of hysteresis effects into drug interactions (e.g., CTZ-MER; Fig. 2B), suggesting that these two phenomena can be mediated by the same molecular mechanisms, as proposed before[19], but that they can also be clearly distinct.

### Negative hysteresis is robustly induced across *P. aeruginosa* strains and populations

The efficient utilization of negative hysteresis as a focus for antibiotic therapy requires that it is robustly expressed across diverse strains of the pathogen. To assess such robustness for clinical antibiotics across a wide panel of strains, we developed a high-throughput OD-based hysteresis screening assay with the five clinically relevant antibiotics MER, PIT, CTZ, GEN, and CIP (Supplementary Fig. S3; Supplementary Data 1 and 2). In our screen we included two reference strains PA14[30] and PAO1[31], the mPact panel covering the genomic diversity of *P. aeruginosa*[25–27], several highly resistant strains from a previous evolu-tion experiment[28], and also diverse *P. aeruginosa* populations newly isolated from COPD patients, thereby yielding a total of around 3.4 million data points and the growth kinetics of over 68,000 cell populations. To account for variation in baseline susceptibility, we adjusted treatment dose to the MIC of each tested sample. By including our PA14 reference strain, we were able to validate the sen-sitivity of our approach to detect both negative hysteresis upon

switches from either β-lactams or CIP to GEN and also positive hys-teresis for the reverse switches (Fig. 3A and Supplementary Fig. S4 and ref. 12), thus indicating that our OD-based screen generates data consistent with the CFU-based assays (Fig. 2A and ref. 12). A Spearman correlation analysis of the CFU to OD relationship using paired data points across the CFU-based hysteresis experiments supported a robust monotonic relationship across pre-treatment antibiotics (Sup-plementary Fig. S5). Consistent with the induction of cell filamentation and slow lysis by β-lactam antibiotics, β-lactam main treatments pro-duced OD values that were higher and more variable than expected from the number of live cells. However, the obtained paired CFU counts and OD values still showed a significant positive correlation ($rho \geq 0.39$, $P \leq 0.0056$; Supplementary Fig. S5). Importantly, the potential confounding factors of OD, such as antibiotic induced cell filamentation or cell death without lysis, are more likely to lead to an underestimation of negative hysteresis (i.e., OD values after pre-treatment are higher than expected from the number of alive cells due to presence of dead cells or filamentation, incorrectly suggesting low cell killing) and an overestimation of positive hysteresis. Therefore, following this reasoning, we consider the OD-based hysteresis screen to provide a conservative indication for the occurrence of negative hysteresis and a potentially biased indication of positive hysteresis, especially when β-lactam antibiotics were used for the main treatment. Accordingly, we focused our following evaluation of the screen's results on the cases showing negative hysteresis.

Our screen of the mPact panel now revealed 121 significant hys-teresis cases across the 412 tested antibiotic switches. Most hysteresis was negative, 81/121, and often associated with a switch from a β-lactam to GEN (Fig. 3A; Supplementary Fig. S4; statistical results in Supplementary Data 3). Importantly, the hysteresis response was not contingent on the strain background and included both strains of clinical and environmental origin (strains H01-H08, H10-H14, and the reference strains PA14 and PA01 are of clinical origin; Supplementary Table S3). These results point to a conserved cellular response mechanism that applies to both clinical and non-clinical strain back-grounds. None of the other switch types produced a similarly robust

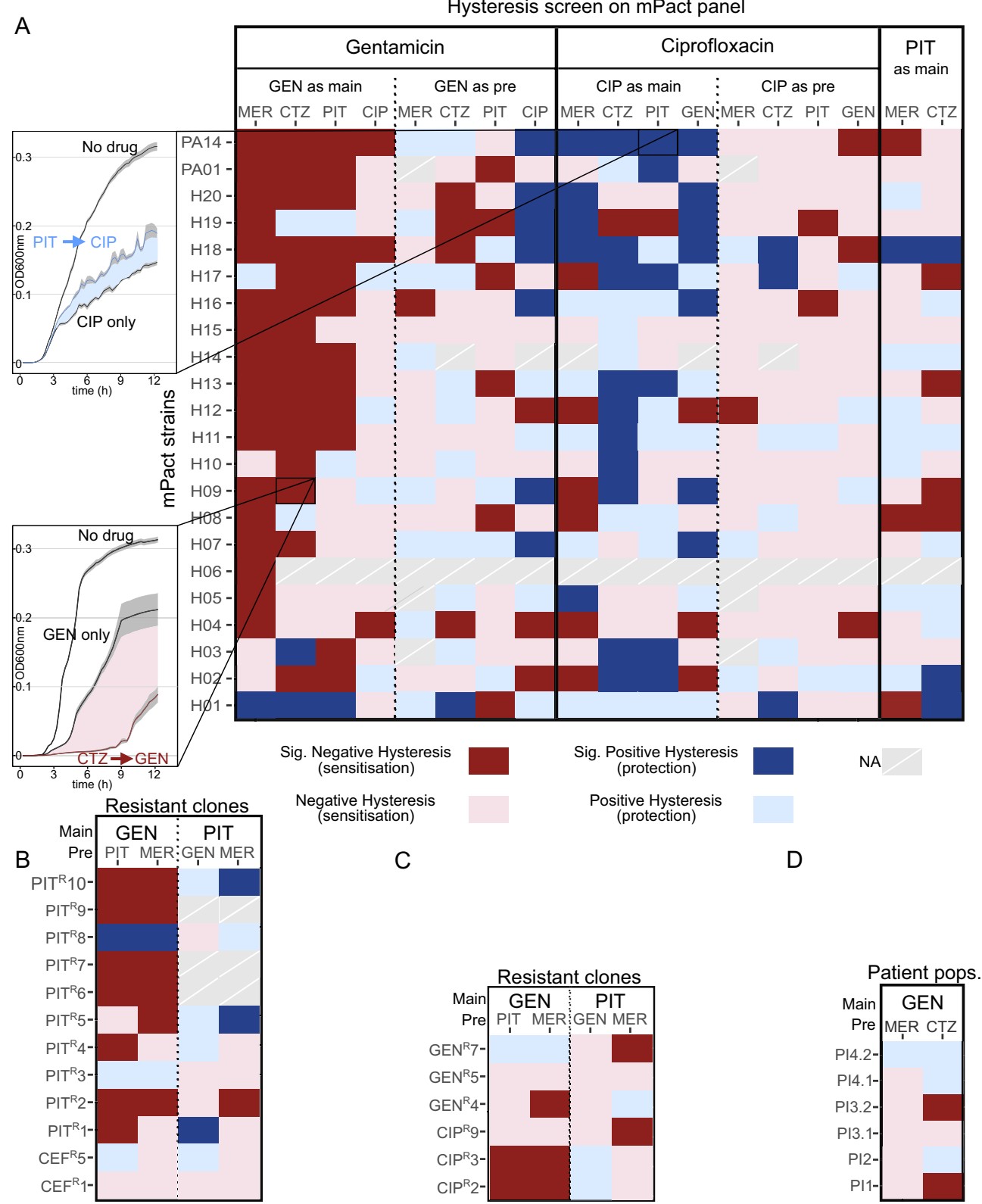

**Fig. legend:**
Sig. Negative Hysteresis (sensitisation) — dark red
Sig. Positive Hysteresis (protection) — dark blue
NA — hatched grey
Negative Hysteresis (sensitisation) — light pink
Positive Hysteresis (protection) — light blue

hysteresis phenotype, suggesting that the other cases of negative hysteresis (Fig. 2A) are mediated through mechanisms that vary across genotypes. Therefore, we focused our subsequent analyses on switches between β-lactams and GEN.

As a next step, we characterized hysteresis in strains resistant to the main treatment antibiotic, the pre-treatment antibiotic, or an unrelated antibiotic. We took advantage of evolved PA14 strains

available from a previous evolution experiment[28] that had evolved high resistance against either GEN, CIP, PIT, or cefsulodin (CEF). Based on our screening assay, we found that 8/10 PIT resistant clones showed negative hysteresis at least once (β-lactam followed by GEN) (Fig. 3B; Supplementary Fig. S4; statistical results in Supplementary Data 3). Strains resistant to the cephalosporin CEF showed no negative hysteresis (Fig. 3B; statistical results in Supplementary Data 3), those

**Fig. 3 | Negative hysteresis is prevalent across the species *P. aeruginosa* upon switches from β-lactam antibiotics to gentamicin.** The hysteresis landscape is visualized as a heatmap. Strains are given on the y-axis, while drug sequences are shown along the x-axis. Significant positive hysteresis appears blue, and significant negative hysteresis red. Non-significant trends ($p > 0.05$) for hysteresis are shown as light blue (for positive) or light red (for negative). We used the clinically relevant antibiotics meropenem (MER), ceftazidime (CTZ), piperacillin/tazobactam (PIT), gentamicin (GEN), and ciprofloxacin (CIP). **A** Hysteresis landscape for the mPact strains. The growth curves on the left depict examples for either positive hysteresis in strain PA14 (PIT followed by CIP) or negative hysteresis in strain H09 (CTZ followed by GEN). Error bands depict mean ± SEM. mPact strains H01-H08, H10-H14, and the two reference strains PA14 and PA01 are of clinical origin, while the remaining strains have an environmental origin. **B** Hysteresis landscape of clones resistant to either PIT (indicated as PIT^R, followed by isolate number) or CEF (indicated as CEF^R). **C** Hysteresis landscape of clones resistant to the main treatment drug GEN (i.e., GEN^R) or to CIP (CIP^R). **D** Hysteresis landscape for diverse strain populations isolated from patients with chronic obstructive pulmonary disease (COPD). Patients (abbreviated PI for patient isolates) are encoded by numbers. Two *P. aeruginosa* populations were considered for patients 3 and 4, isolated at different time points. Each patient population consisted of 8–14 isolates. For all statistical comparisons in (**A**–**D**), we used two-sided pair-wise Wilcoxon rank sum tests and adjustment of $p$ values by the false discovery rate to assess significant differences between respective hysteresis and control treatments; each comparison (i.e., each tile of the heatmap) was based on 6–14 technical replicates. The detailed statistical test results are given in Supplementary Data 3. The data for this figure is provided in the supplementary Source data file.

resistant to CIP showed negative hysteresis in 2/3 cases (Fig. 3C; statistical results in Supplementary Data 3), while the GEN-resistant strains only showed negative hysteresis in a single case (Fig. 3C; Supplementary Fig. S4; statistical results in Supplementary Data 3). Overall, these results indicate that with MIC-adjusted pre-treatments, negative hysteresis is induced in a number of antibiotic-resistant strains, in particular those expressing resistance to the sensitizing pre-treatment antibiotic.

Recognizing that chronic *P. aeruginosa* infections in patients may involve several genotypes[32,33], we further examined hysteresis in diverse *P. aeruginosa* populations, where the strains of each population had been isolated from sputum samples of individual COPD patients. We reconstructed six mixed-strains *P. aeruginosa* populations from four patients (two patients produced positive sampling cultures on different days). The individual patient isolates were initially cultured separately and then combined at equal ratios to reconstruct each respective within-patient population, with each population comprising 8–14 phenotypic isolates. When each was subjected to the hysteresis assay, we found that two of the reconstructed populations showed significant negative hysteresis upon switches from CTZ to GEN. Most switches increased treatment efficacy, none were inhibitory (i.e., there were no cases of positive hysteresis; Fig. 3D; Supplementary Fig. S4; statistical results in Supplementary Data 3). These proof-of-concept findings indicate that negative hysteresis is inducible in a diverse pathogen population, representative of chronic infections.

## The Cpx envelope stress response system contributes to antibiotic-induced negative hysteresis

Next, we examined which molecular mechanism(s) underpin negative hysteresis. Considering the previous implication of a particular *cpxS* mutation in hysteresis[12], we focused our analysis on the Cpx (conjugative pilus expression) two-component, envelope stress response system and its involvement in the β-lactam-induced sensitization of bacterial cells. The Cpx system is well characterized in *Escherichia coli*, where it regulates inner membrane integrity and efflux pumps[34,35], influences the susceptibility towards β-lactams and aminoglycosides[36–39], and mediates expression of virulence factors, the latter additionally shown for other Gram-negative bacteria (reviewed in ref. 40). The distinct Cpx system of *P. aeruginosa* is currently getting increasing attention[41–44]. The left panel of Fig. 4A illustrates the *P. aeruginosa* Cpx system, highlighting the structural differences in the sensor histidine kinase CpxS with the *E. coli* CpxA kinase[12]. In detail, while the signaling domain (HAMP) and catalytic domains (DHp and CA) superimpose well (Supplementary Fig. S6), the CpxS sensory domain (residues 8–148) is clearly distinct from that of *E. coli* CpxA. Thus, we used the sensory domain of AbfS, a sensory histidine kinase from *Cellvibrio japonicus* (PDB ID: 2VA0)[45], as an alternative template for structural modeling of the sensory domain (right panel, Fig. 4A). In analogy with the Cpx system of *E. coli*, we assume that in *P. aeruginosa*, various stressors induce autophosphorylation of CpxS, which in turn

phosphorylates the transcription factor CpxR. CpxR should activate – among others – CpxP, which itself should inhibit CpxS in a negative feedback mechanism. Based on this model, we generated mutants for different components of the Cpx system.

As a first step, we re-assessed the previously identified contribution of the specific CpxS T163P mutation to negative hysteresis[12] and confirmed that the mutation consistently abrogated the CAR-mediated sensitization of cells towards GEN in pure and mixed cultures with different ratios of the mutant and wildtype (Fig. 4B; Supplementary Fig. S7). Thereafter, we characterized scar-free deletion mutants of the three Cpx system genes (*cpxS, cpxR, cpxP*) and two additional *cpxS* mutants with specific mutations (CpxS DES81–83G and CpxS G188S) originally observed in the previous evolution experiment[12]. We always used mutant-adjusted antibiotic concentrations for the hysteresis treatment (0.5 MIC for CAR pre-treatment, 1 MIC for GEN main treatment; Supplementary Table S4) and included PA14 wildtype (WT) as positive and CpxS T163P as negative controls. The results are summarized in Fig. 4C as a heatmap, inferred as above via the ΔAUC between the hysteresis and control treatments (each heatmap tile represents the data of at least six time-kill curves). We found that deletion of *cpxR* abrogated the WT hysteresis phenotype, whereas deletion of *cpxS* and *cpxP* did not (Supplementary Fig. S8A; statistical results in Supplementary Table S5). The three considered *cpxS* mutations (T163P, DES81-83G, G188S) consistently abolished the hysteresis phenotype present in the Δ*cpxS* mutant and WT, suggesting that they are CpxS gain-of-function mutants. These gain-of-functions are likely mediated by an increased activity of CpxS. This idea was confirmed by analysis of additional mutants with an inducible CpxS overexpression system in WT as well as *cpxS* deletion background, for which the induced overexpression of CpxS led to a loss of significant negative hysteresis (Fig. 4D; Supplementary Fig. S8B; statistical results in Supplementary Table S5). Moreover, we found that the loss of negative hysteresis in the CpxS T163P mutant was associated with a significant change of drug interaction profile, whereby WT-level synergistic interactions between both CAR-CIP and CAR-GEN consistently shifted towards an additive interaction in the mutant (Fig. 4E; Supplementary Fig. S9; statistical results in Supplementary Table S6).

To further characterize the molecular underpinnings of CpxS-mediated effects on negative hysteresis, we performed a transcriptomic analysis of WT PA14 and the CpxS T163P mutant with or without CAR pre-treatment (50 mg/L CAR for 30 min). A comparison between pre-treated and untreated WT identified *flgB* as the only significantly differentially expressed gene (absolute fold-change ≥ 2; down-regulated; statistical results in Supplementary Data 4 and 5). More interestingly, the comparison of the gain-of-function CpxS T163P mutation with the WT revealed 124 significantly differentially expressed genes (absolute fold-change ≥ 2; Fig. 4F; statistical results in Supplementary Data 6 and 7) that include genes involved in transmembrane transport, type III secretion, quorum sensing, and

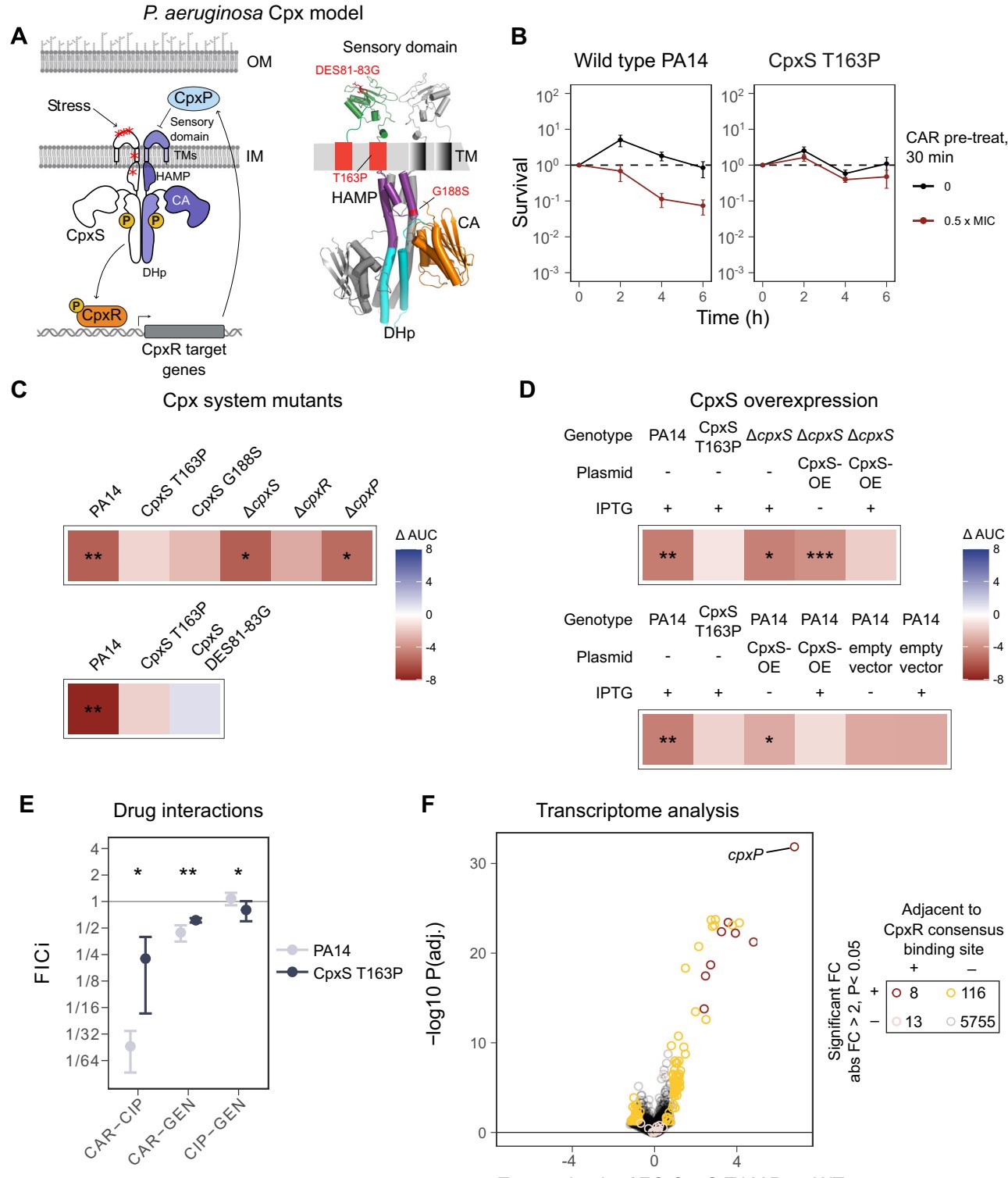

**A** *P. aeruginosa* Cpx model

**B** Wild type PA14 / CpxS T163P — Survival vs Time (h); CAR pre-treat, 30 min (0, 0.5 × MIC)

**C** Cpx system mutants

**D** CpxS overexpression

**E** Drug interactions

**F** Transcriptome analysis

pathogenesis, suggesting that CpxS activation in the mutant affects membrane structure and membrane-associated functions (Supplementary Data 5, 6, and 7). In addition, eight of the 124 genes are located close to a consensus CpxR binding site[46], identifying them as the core set of the Cpx regulon, including *cpxP* as the most strongly upregulated gene (more than 100-fold) in CpxS T163P, supporting the role of CpxP in a feedback loop with CpxS activation (statistical results in Supplementary Data 7). Consistent with our findings, *cpxP* is also the most positively CpxR-regulated gene in *E. coli*[47,48]. Furthermore, *cpxS* itself is strongly upregulated in the CpxS T163P mutant, again

supporting the presence of feedback loops in regulating this gene (statistical results in Supplementary Data 7).

**Negative hysteresis is associated with changes in membrane stress response and increased uptake of gentamicin**

To further examine the mechanisms underlying negative hysteresis, we investigated which cellular changes are mediated by the Cpx envelope stress response system. Considering that the most effectively sensitizing antibiotics (i.e., β-lactam drugs) target cell wall synthesis, that CpxS should be localized in the inner membrane (in analogy with

**Fig. 4 | The Cpx envelope stress response system is involved in the expression of negative hysteresis. A** Proposed overview of the Cpx system of *P. aeruginosa* (left) based on that of *E. coli*. Red asterisks indicate the location of *cpxS* gene mutations that were previously observed during experimental evolution[12]. The 3D homology model (right) was generated based on the structure of *E. coli* CpxA (intracellular part)[62] and the AbfS sensor domain of *Cellvibrio japonicus*[45]. The mutations highlighted in red were further characterized in this study (Membrane and double helix icons: Created in *BioRender*. Upterworth, L. (2026) https://BioRender.com/96iv8ws). **B** Exemplary CAR-GEN hysteresis time-kill curves for the wild-type PA14 (positive control in **C**, **D**) and the CpxS T163P mutant (negative control in **C**, **D**) (*n* = 3 per treatment and strain; Error bands depict mean ± SEM). **C** Heatmaps show the results of Carbenicillin (CAR) – Gentamicin (GEN) hysteresis time-kill experiments as mean area under the curve (ΔAUC) for deletion mutants of all three components of the Cpx system (*cpxS*, *cpxR*, *cpxP*) and two strains with mutations in *cpxS* (DES81-83G, G188S). Red indicates negative hysteresis, blue positive hysteresis, and asterisks significant negative hysteresis, based on two-sided Student's *t*-test (*P < 0.05; **P < 0.01; ***P < 0.001; top panel: *n* = 3 per treatment and strain; bottom panel: *n* = 5 per treatment and strain). **D** Summary of CAR-GEN hysteresis time-kill experiments with IPTG-inducible CpxS overexpression

constructs in different strain backgrounds (PA14, Δ*cpxS*). IPTG was added to control strains to account for possible IPTG side effects (Two-sided Student's *t*-test, *n* = 5 per treatment and strain). **E** Drug interaction profiles for PA14 and CpxS T163P obtained by CombiANT assays and shown as fractional inhibitory concentration index (synergy: FICi < 0.5; antagonism: FICi > 4; additive interaction: 0.5 < FICi < 4) Error bands depict mean ± SD. Drug pairs with CAR are less synergistic in CpxS T163P (Two-sided Wilcoxon rank sum test: *P*-values rounded to 3 digits after comma from left to right *p* = 0.032, *p* = 0.008, *p* = 0.032; *n* = 5). **F** Volcano plot comparing the mean gene expression across three time points (0, 30, and 60 min) of PA14 and CpxS T163P (*n* = 4 per treatment and strain). The x-axis depicts the fold-change in gene expression compared to PA14 and the y-axis the FDR-adjusted *P*-values for each of the genes, inferred with the quasi-likelihood generalized negative binomial model and associated F tests, as implemented in *edgeR*[95]. Genes highlighted in color show a significant fold-change as well as an absolute fold-change > 2. Genes that are located near a CpxR consensus binding site are additionally highlighted in red. The data for this figure is provided in the supplementary Source data file. The detailed statistical test results are given in Supplementary Tables S5, S6, and Supplementary Data 7.

what is known for the *E. coli* ortholog CpxA), and that the Cpx system responds to membrane stress[34–36], we postulated that disruption of cell envelope integrity and/or increased permeability could lead to negative hysteresis. We tested these ideas with the help of three envelope stressors as pre-treatment: (i) polymyxin B (PMB), which targets the outer cell membrane, (ii) CAR, which targets the peptidoglycan matrix, and (iii) benzyl alcohol (BnOH), a general membrane fluidizer[49]. We found that negative hysteresis is induced in the WT PA14 by both CAR and BnOH but not PMB, while the CpxS T163P mutation inhibits WT-level sensitization by CAR (as shown above) but not BnOH (Fig. 5A; statistical results in Supplementary Table S7). These results indicate the involvement of the inner membrane and the peptidoglycan layer in negative hysteresis. Since the CpxS T163P mutant did not inhibit BnOH-mediated sensitization, BnOH may induce negative hysteresis differently or even independently from peptidoglycan-targeted hysteresis.

We then asked how CAR pre-treatment affects membrane stress responses in WT PA14 and the CpxS T163P mutant using Laurdan, a fluorescent membrane dye and sensor for lipid packing that produces a spectral shift towards lower wavelengths with decreasing membrane fluidity[50,51]. We found that CAR exposure led to a significant membrane stress response in the WT (Fig. 5B; statistical results in Supplementary Table S8). Thus, the spectral shift indicated CAR-induced membrane rigidification during early growth at the used sub-lethal concentration. In contrast, the active membrane stress response in the CpxS T163P mutant already produced a rigidified membrane, which was then not significantly affected by an additional CAR pre-treatment (Fig. 5B; Supplementary Table S8). These findings strongly suggest that membrane pre-adaptation via constitutive CpxS expression actively suppressed hysteresis in the mutant. An orthogonal way to modify membrane fluidity are sudden shifts in temperature, to which bacteria respond by fluidity-adjusting membrane remodeling (e.g., see ref. [49]). We thus reasoned that a heat-shock could provide an equivalent stimulus as hysteresis-inducing CAR pre-treatments. As a test, we induced a heat shock by a sudden up-shift of temperature from 37 °C to 50 °C for 15 min, ahead of the pre-treatment step. Consistent with the membrane stress hypothesis, we expected (i) the heat shock by itself to accelerate GEN killing and (ii) the heat shock to suppress the possibly redundant inducing effect of CAR. The experiment confirmed both predictions (Fig. 5C, statistical results in Supplementary Table S9), supporting our model that abrupt changes to the membrane fluidity trigger the CAR-induced GEN hysteresis. As low permeability of the inner membrane limits aminoglycoside uptake in Gram-negatives[52], we hypothesized that membrane stress and dynamic membrane remodeling altered aminoglycoside membrane

permeation, leading to the observed accelerated lethality of hysteresis. The uptake of aminoglycoside is energized by the proton motive force of the cytoplasmic membrane[53]. Thus, induced dynamics of proton motive force may link pre-treatment inhibition to hysteresis lethality. The chemical inhibitor CCCP allows protons to cross the cytoplasmic membrane, collapsing membrane potential such that no dynamic change of the membrane is possible[54]. When we pre-incubated cells with 200 µM CCCP, cell numbers did not increase, relative to controls treated with solvent only. This reduction is likely explained by a requirement of membrane potential for growth. The pre-treatment with only CAR was more effective than CCCP at inducing negative hysteresis. Intriguingly, the CCCP incubation precluded any additional effects of CAR pre-treatment on hysteresis (Fig. 5D, statistical results in Supplementary Table S10), further supporting the idea that an altered cell envelope underpins negative hysteresis.

We finally tested to what extent β-lactam-mediated sensitization towards GEN is directly related to an increased uptake and thereby intracellular abundance of the latter drug. Using an ELISA assay for GEN in our hysteresis assay with or without the pre-treatment, we observed a significantly increased intracellular GEN concentration upon CAR pre-treatment in the WT PA14 but not the CpxS T163P mutant (Fig. 5E, statistical results in Supplementary Tables S11 and S12). β-lactam induced negative hysteresis to GEN is thus explained by increased uptake of the aminoglycoside resulting from Cpx-dependent physiological remodeling of the inner cell membrane.

## Discussion

Our results yield fundamental insights into the phenomenon of negative hysteresis and the drug-induced sensitization of bacterial cells towards specific antibiotics in the high-risk pathogen *P. aeruginosa*. The exact characteristics of this phenomenon were previously unknown. Thus, it was unclear which exact conditions and switching directions lead to cellular sensitization, how robust is the effect across the genomic diversity of a pathogen taxon, whether it can be induced in resistant strains or mixed-strain pathogen populations, and which mechanisms contribute to sensitization. Our study now demonstrates that negative hysteresis is robustly induced upon switches from β-lactam antibiotics to the aminoglycoside gentamicin (Figs. 1–3). Negative hysteresis represents a conserved physiological response in *P. aeruginosa*, which is inducible by sub-MIC doses of the sensitizing antibiotic (Figs. 1B, 4B–D, 5) and which can be elicited in antibiotic resistant *P. aeruginosa* mutants as well as strain mixtures isolated from infected patients (Fig. 3D). The robustness of the response, even at sub-MIC levels of the sensitizing antibiotic, highlights the potential of

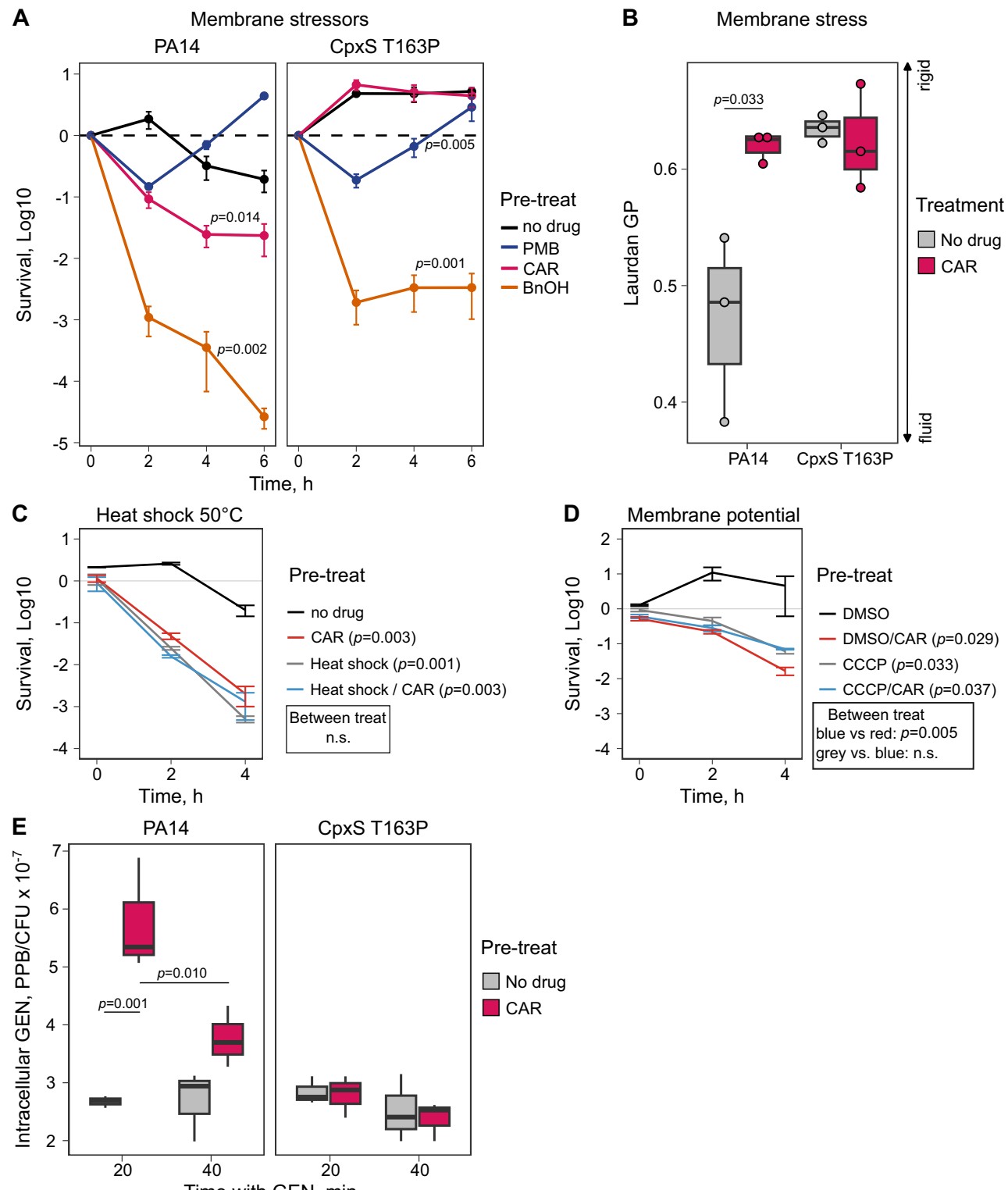

negative hysteresis to transiently suppress intrinsic resistance and thereby improve antibiotic therapy. Furthermore, the robustness of hysteresis across *P. aeruginosa* diversity eliminates the need for specific prior phenotypic tests. The transience of hysteresis (Fig. 1G), however, ideally requires a gapless antibiotic sequence, or their staggered application.

What is the most likely explanation for the observed negative hysteresis? One simple mechanism to explain negative hysteresis is that unbound pre-treatment antibiotic is retained in the treated cells

(intracellularly or in the periplasmic space), which then contributes to killing in the main treatment. However, the washing experiment (Supplementary Fig. S2B) indicated that these effects play a negligible role for the case of CAR-GEN hysteresis, possibly because β-lactam effects are very dose-sensitive around the MIC (producing steep dose response curves) so that residual amounts are expected to be ineffective. Moreover, MexR mutants that overproduce MexAB-OprM efflux pumps show hysteresis[12], although such mutants efflux unbound β-lactam efficiently.

**Fig. 5 | β-lactam-induced negative hysteresis is associated with changes in membrane characteristics and increased gentamicin uptake.** All assays were performed with the wildtype PA14 and the CpxS T163P gain-of-function mutant. **A** Time-kill curves for the main treatment gentamicin (GEN) and a pre-treatment with agents affecting the inner and outer cell membranes, including polymyxin B (2 mg/L; PMB) for the outer membrane, carbenicillin (50 mg/L; CAR) disrupting peptidoglycan synthesis, and benzyl alcohol (0.4 %; BnOH) impacting both membranes. When only the outer membrane was targeted, there was no hysteresis in PA14. Statistical difference to the no-drug control was inferred by two-sided Student's *t*-test based on area under the curve (AUC), corrected for multiple testing using the false discovery rate (FDR) (mean ± SEM, *n* = 3). **B** Laurdan fluorometric measurement of membrane stress. After a 30 min CAR treatment 50 mg/L, the Laurdan generalized polarization (GP) for both strains was measured. CAR induced membrane stress for PA14. CpxS T163P has inherently elevated levels, which were not affected by CAR treatment. The significance of a difference between hysteresis and control treatment was assessed with a two-sample *t*-test (*n* = 3). **C** Induction of a mild heat shock (15 min 50 °C) functionally replaces a 30 min CAR pre-treatment of

50 mg/L. Heat-induced cells do not show hysteresis, while control treatments do. Statistical difference was assessed by comparing the survival at 4 h with a two-sided Student's *t*-test and correcting for multiple comparisons by FDR (mean ± SEM, *n* = 3). **D** Collapse of membrane potential by 0.2 mM CCCP for 15 min suppresses additional effects of a 30 min CAR pre-treatment. CCCP-treated cells do not show hysteresis, while control treatments with DMSO solvent show hysteresis. Statistical difference was assessed by comparing the survival at 4 h with a two-sided Student's *t*-test and correcting for multiple comparisons by FDR (mean ± SEM, *n* = 3). **E** Intracellular GEN concentrations determined by ELISA 20 and 40 min after GEN addition upon CAR hysteresis or control treatment. CAR pre-treatment increased intracellular GEN concentrations only in PA14 but not in CpxS T163P. Boxplots in (**B**, **E**) display the median (line), the first and third quartiles (box edges), and whiskers extending to the smallest and largest values within 1.5× the interquartile range. Significant differences were evaluated with an ANOVA, followed by pairwise comparisons using Tukey's HSD tests with FDR correction (*n* = 3). The data for this figure is provided in the supplementary Source data file. The detailed statistical test results are given in Supplementary Tables S7,S8, S9, S10, S11, and S12.

An alternative explanation is that the pre-treatment antibiotic induces a cell envelope damage that influences the effect of the second drug. CAR and other β-lactams covalently bind their target penicillin binding proteins (PBPs), weakening the cell wall structure by preventing peptidoglycan cross-links[55]. This damage could be carried into the main treatment, either through the retention of the covalently bound β-lactam and/or the maintenance of the resulting effect of compromised cross-links between peptidoglycan molecules. Either of these alternatives may provide a simple and robust memory mechanism that is diluted by PBPs newly formed during later growth. In addition, based on the current evidence, alternative memory forms such as protein aggregates[56] or response memory[57] cannot yet be excluded. The experiments with double pre-treatments (50 °C, chloramphenicol, CCCP) showed that the induction of negative hysteresis is related to membrane dynamics during mild heat stress and that the effect is biologically controlled (Fig. 1E and Fig. 5C, D). Hysteresis requires growth to establish, as it can be blocked by an inhibitory dose of chloramphenicol. The similar ability of CCCP to block the build-up of hysteresis, may stem from either growth inhibition (proton motive force is required for growth), inhibited antibiotic entry or their combination. These observations are consistent with the idea that hysteresis builds up through dynamic changes to the cell envelope, which are likely induced by β-lactam drugs covalently bound to PBP and the resulting reduction in cross-linked peptidoglycan and/or other β-lactam-like stress. Furthermore, our results also suggest that negative hysteresis is most likely caused by the pre-treatment induced increase in the uptake of the main treatment drug (Fig. 5E).

Our results demonstrate a central role of the Cpx cell envelope stress response system, prevalent across Gram-negative bacteria[36,58], in mediating hysteresis. β-lactam antibiotics are known to disrupt peptidoglycan synthesis, thereby directly affecting cell envelope properties, which can be sensed by the Cpx system in *E. coli*, leading to its activation[59]. In turn, Cpx activation itself influences membrane integrity, as previously shown in various Gram-negative bacteria[60,61]. Based on these considerations, we propose that β-lactam pre-treatment destabilizes the cytoplasmatic membrane (Fig. 5A, B), which induces a membrane stress response that is regulated by the intact Cpx system of *Pseudomonas* (Figs. 4F, 5B). We further propose that high levels of negative hysteresis selectively favor CpxS gain-of-function mutations to counter membrane disruption. These suggestions are supported by our current and previous results. Thus, the tested *cpxS* mutations were selectively favored by sequential treatments with high levels of negative hysteresis during a previous evolution experiment[12]. CpxS gain-of-function mutations lead to a constitutively active CpxS, which inhibits the sensitization of bacterial cells upon β-lactam-aminoglycoside

switches (i.e., abrogation of negative hysteresis; Fig. 4B–D), thereby reducing the uptake of the aminoglycoside antibiotic gentamicin (Fig. 5E). The reduced gentamicin uptake in the CpxS T163P mutant is likely to be linked to increased activity of the MexAB-OprM efflux pump, as observed in the transcriptomics analysis (Fig. 4F, statistical results in Supplementary Data 6 and 7), a notion consistent with recent functional and transcriptomic analyses of the same CpxS T163P mutation[41,42,44]. Since the three *cpxS* mutations, now characterized by us, are located in different domains of the protein (Fig. 4A), they likely modulate CpxS activity differently, resulting in the observed variations in negative hysteresis abrogation (Fig. 4C). Previous work in *Enterobacteriaceae* showed that mutations in the sensor domain and second transmembrane domain of CpxA cause CpxA overactivation and a disrupted interaction with CpxP[35,60]. This also appears to apply to *cpxS* mutations in *P. aeruginosa*[41]. An increased Cpx activity is linked to increased antibiotic resistance at low nutrient levels[41]. While the T163P mutation does not cause clear resistance in our media (change of MIC <2x), it does increase antibiotic tolerance[12]. The T163P mutation in the transmembrane domain has the potential to modify the orientation of the sensory domain, while the DES81-83G is located in the sensory domain, possibly affecting its conformation. Both changes may alter signal processing of the sensory domain and/or its interaction with CpxP, leading to a continuously activated CpxS. Furthermore, *E. coli* CpxA activation is modulated by a helix twist in the HAMP domain[62]. In our case, the CpxS G188S mutation is located within the HAMP domain in one of the two helices involved in this twist (Fig. 4A; Supplementary Fig. S6). This could conceivably affect structural organization of the HAMP domain and ultimately cause CpxS activation. Intriguingly, heat-shock does not sensitize *E. coli* to aminoglycoside in sequential designs although it does when heat-shock and aminoglycoside treatment co-occur[63]. One possible explanation for these observations is that the structural differences in the sensory kinase encode a simple regulatory memory of heat-shock-like damage, such as misfolded proteins, in *P. aeruginosa*. Such memory could additionally stem from interactions with the recently identified CpxM and CpxH modulator proteins of the *Pseudomonas* Cpx response[44].

Mutations in *cpxS* were found in evolution experiments[43,46,64,65] as well as in multi-drug resistant clinical isolates[43,66]. The T163P mutation seems to be particularly advantageous under β-lactam selection and has been observed for a variety of β-lactam treatments in different media and multiple strain backgrounds[12,28,41,42,67–69]. These observations suggest that Cpx is a prevalent target of the selection imposed by antibiotic treatment, especially β-lactams, which itself may then be associated with antibiotic-induced cell stress and cellular sensitization. Furthermore, *cpxS* mutations were found to modulate pyoverdine production[42] and antibodies against CpxS were found in CF patient

sera[70] indicating that the Cpx system of *P. aeruginosa* also regulates virulence and may therefore be subject to evolutionary trade-offs. Very recently, the *Pseudomonas* Cpx system was found to sense surface attachment and induce iron scavenging during biofilm formation[44]. Overall, these observations indicate a clinical relevance of genetic variation in *cpxS*[71], further highlighting the need for a better understanding of this stress response system.

We were surprised to find that negative hysteresis can be induced in mutants that are resistant to the pre-treatment drug (Fig. 3B, C). Within a certain range, sub-inhibitory concentrations induce hysteresis. Usually, an antibiotic would not be considered for a strain that is resistant against this drug. Negative hysteresis may therefore provide the unique opportunity for re-introducing β-lactams into treatment regimens that otherwise may have been deemed ineffective. Such repurposing of antibiotics (reviewed in ref. 72) is, however, limited by the necessity to use MIC-adjusted concentrations, which may exceed drug concentrations that can be applied to patients. Our results further highlight that the induction of negative hysteresis is only possible in case of resistance to the pre-treatment drug, but not when bacteria are already resistant to the main treatment antibiotic GEN. Moreover, negative hysteresis may also be abrogated by specific resistance mutations that are related to the cell envelope, as previously identified for the two tested CEF-resistant strains[28], for which we could not identify any β-lactam-mediated sensitization.

Our findings further suggest that negative hysteresis is related, although not identical to the molecular processes underlying drug interaction synergy, as proposed previously[19]. There is overlap, albeit not complete, in the antibiotic pairs causing negative hysteresis and synergy (Fig. 2). As observed before, several synergistic drug pairs (e.g., CTZ-GEN), produce a mild positive hysteresis in addition to the expected strong negative hysteresis. The sequential experiment thus provides evidence for single-sided synergy induction and inducible protective responses (Fig. 2A, B). While the interaction profile could be inferred from hysteresis measurements for the CTZ-GEN pair, another example in our data shows that this is not always so: switches between CTZ and meropenem (MER) consistently produce positive hysteresis, but their combination shows synergy (Fig. 2A, B). These examples underscore that, while there is a relationship between negative hysteresis and synergy, the two phenomena are distinct. For interactions with GEN, the phenomena are clearly related, as the studied, CpxS T163P mutation abrogates cellular sensitization (Fig. 4B–D) and simultaneously causes a reduction of synergy in antibiotic interactions (Fig. 4E). This connection between antibiotic cellular sensitization and synergistic drug interactions is consistent with earlier data from 1962 on the cellular underpinnings of β-lactam – aminoglycoside synergy (i.e., synergy between penicillin and streptomycin) in *E. coli*[20].

In conclusion, our findings show the potential of β-lactam antibiotics to induce robust sensitization of bacterial cells towards aminoglycoside antibiotics in *P. aeruginosa*. Such negative hysteresis may also be involved in the antiseptic-induced transient sensitivity towards the aminoglycoside tobramycin, previously described for *P. aeruginosa*[73]. Importantly, negative hysteresis is likely expressed by other bacterial taxa as well, as already now strongly indicated by the above study on penicillin-streptomycin synergy in *E. coli* ML35[20], the previously reported increase of daptomycin efficacy in *Staphylococcus aureus* following β-lactam exposure[23,74], and the reduction of persistence levels by specific staggered applications of bacteriostatic and bactericidal antibiotics in *E. coli* and *S. aureus*[22]. It likely also underlies the previously published increase in *P. aeruginosa* eradication and health improvements in a cohort of infected CF patients, who were subjected to a sequential treatment of first a β-lactam followed by an aminoglycoside drug[75]. These latter clinical results at least suggest that negative hysteresis could be valuable as a focus for optimizing antimicrobial treatment in vivo.

## Methods

### Ethics statement
Our study included clinical bacterial isolates collected from sputum samples of patients. These patients were enrolled in the clinical observational trial "Airway colonization with *Pseudomonas aeruginosa* in chronic obstructive pulmonary disease (COPD)/non-CF bronchiectasis - an observational and biomaterial study" at LungenClinic Großhansdorf (Großhansdorf, Germany), approved by the local ethics committee of the Medical Faculty of the University of Lübeck (No. 20-295), available through the German Clinical Trials Register (https://drks.de/, ID DRKS00023975). Prior to enrollment, patients provided written informed consent for participation.

### Bacterial strains, culture conditions, and antibiotics
An overview of all bacterial strains, plasmids, and antibiotics used or generated in this project is provided in the Supplementary Data 8, 9, and 10. We used 6 patient populations from four patients all including 8–14 isolates, two patients had two sample time points included, that were collected between January and November 2021. Only three of these samples had additional bacterial species isolated (Supplementary Table S13), whereby we here focused solely on *P. aeruginosa* as a model pathogen. Further information on the individual isolates, such as morphology and MIC values of the patients, can be found in the Supplementary Data 11.

### Antimicrobial susceptibility testing
Antimicrobial susceptibility was assessed by determining the minimum inhibitory concentration (MIC) of an antibiotic, following three main approaches: (i) a broth microdilution approach, (ii) the Vitek®2 (bioMérieux) diagnostic system (always following manufacturer instructions), and (iii) MIC test strips (Liofilchem®, following manufacturer instructions), depending on the respectively required resolution. The MIC determination used was consistent within each panel. The broth microdilution was based on the EUCAST (European Committee for Antimicrobial Susceptibility Testing) guidelines for antimicrobial susceptibility testing[76], using 96-well plates and a 2-fold serial dilution of the antibiotic, incubation statically at 37 °C for approximately 20 h, and measurement of optical density at 600 nm ($OD_{600}$). MIC was determined as the lowest concentration that resulted in a relative OD (OD relative to growth control) below 0.1.

### Hysteresis time-kill assay
Hysteresis time kill assays were used to measure how pre-treatment with one antibiotic affected killing efficacy of a subsequently administered antibiotic. The assays were initiated with a bacterial culture at early exponential phase in M9 medium, which was subjected to a pre-treatment at 37 °C with the antibiotic of interest for up to 30 min, as indicated. Thereafter, the pre-treatment antibiotic was removed by centrifugation. The pellet was resuspended immediately in fresh medium and the main treatment drug was added, as indicated. To assess the persistence of the pre-treatment-induced effect, we introduced a drug-free period of up to 80 min before initiating the main treatment. The samples were then incubated at 37 °C and bacterial survival evaluated as colony forming units (CFU) per mL culture after 0, 2, 4, and 6 h of main treatment, following standard protocols[12]. The area under the curve (AUC) of the obtained time-kill curves was calculated based on the CFU or survival data. The hysteresis effect was quantified as the difference in AUC (ΔAUC) between the hysteresis and the corresponding control treatment and summarized in a heatmap. The AUC data were also used in some cases to generate a Nelder-Mead dose-response model[77] of the hysteresis effect size. The significance of differences between hysteresis and control treatments was assessed using a two-sided Student's *t*-test. The relationship between CFU per mL and OD was assessed by non-parametric Spearman's rank sum correlation of paired CFU and OD data points from the 6 h timepoint.

The correlation was performed separately for main treatments with β-lactam and for measurements of OD by cuvette or microtiter plate (Supplementary Fig. S5).

## Hysteresis wash experiment

To test for a possible role of antibiotic carry-over in hysteresis induction we performed a standard hysteresis experiment (PA14, pre-treatment CAR 50 mg/L 30 min) but inoculated two cultures from each pre-culture so that one group could be treated as usual (with regular carry-over of pre-treatment CAR) and one group washed thoroughly with phosphate buffered saline to remove free and thus unbound CAR (two washes with 5 mL PBS, pellet cells by centrifugation 10 min at 4000 g, 4 °C) before start of the GEN treatment 1 mg/L. Error bars show SEM for $n = 3$ biological replicates. A two-sided Student's $t$-test was used for statistical testing. This treatment will not remove any covalently bound CAR.

## Hysteresis experiments with complex pre-treatments

a) Chloramphenicol: To test whether active growth was required for hysteresis, we diluted a dense overnight culture to OD 0.005 in fresh medium and grew it to OD of 0.06-0.07. We then split the growing culture into parallel 5 mL cultures and added 512 mg/L chloramphenicol (8x MIC, stock prepared in ethanol 95%) to stop protein production or an equivalent volume of the solvent (final ethanol concentration in culture 3%). After 15 min, we added CAR 50 mg/L (0.5x MIC) to initiate pre-treatment. After 30 min, cells were pelleted by centrifugation and resuspended in medium containing 1 mg/L GEN (1 x MIC) as the only antibiotic. Survival during GEN treatment was measured using CFU as described in the hysteresis standard protocol. Error bars show SEM for $n = 3$ biological replicates.

b) Heat shock: To test whether a mild heat shock could functionally replace a CAR pre-treatment, we grew cells from OD of 0.005 (diluted overnight culture) to 0.05 and induced heat shock by shifting half of the culture from 37 °C to a heated water bath of 50 °C for 15 min before splitting the culture into parallel 5 mL cultures and adding 50 mg/L CAR (0.5 x MIC) for hysteresis induction. After 30 min pre-treatment with CAR, cells were pelleted by centrifugation and resuspended in medium containing 1 mg/L GEN (1 x MIC) as the only antibiotic. Survival during GEN treatment was measured using CFU as described in the hysteresis standard protocol. Error bars show SEM for $n = 3$ parallel cultures. Statistical analysis was performed by comparing the survival after 4 h (Two-sided Student's $t$-test with FDR correction).

c) Membrane potential: To test the role of membrane potential in negative hysteresis, we grew cells from OD of 0.005 (diluted overnight culture) to 0.06 and collapsed membrane potential by adding 0.2 mM carbonyl cyanide 3-chlorophenylhydrazone (CCCP, stock prepared in DMSO) or an equivalent amount of the solvent (final concentration of DMSO in culture was 0.8%). After 15 min, we added CAR 50 mg/L (0.5x MIC) to initiate pre-treatment. After 30 min, cells were pelleted by centrifugation and resuspended in medium containing 1 mg/L GEN (1 x MIC) as the only antibiotic. Survival during GEN treatment was measured using CFU as described in the hysteresis standard protocol. Error bars show SEM for $n = 3$ biological replicates. Statistical analysis was performed by comparing the survival after 4 h (Two-sided Student's $t$-test with FDR correction).

## Hysteresis screen experimental design and protocol

We developed a high-throughput screen to assess the presence or absence of hysteresis across a larger number of antibiotics and *P. aeruginosa* strains. The assay was based on continuous OD-measurements, taken in plate readers (Epoch2, Agilent) in 15-min intervals across a 12 h time period, to infer growth kinetics of bacterial cultures in 384-well plates, which were subjected to hysteresis or corresponding control treatments (i.e., either with or without antibiotic pre-treatment, respectively). Drug concentrations were standardized for each bacterial strain and antibiotic, using information on MIC inferred with the Vitek®2 (bioMérieux) approach (Supplementary Data 1). We used two concentrations for the pre-treatment at 0.75 and 0.075 MIC, and four concentrations for the main treatment at 0.75, 0.375, 0.075, and 0.0375 MIC.

For data analysis, we focused on pre-treatment drug concentrations that did not directly affect growth. Samples were excluded from the statistical analysis, if both used pre-treatment drug concentrations directly inhibited growth. To account for the unidirectional expectation, a one-sided Wilcoxon-rank-sum test adjusted for multiple comparison with the false discovery rate (FDR) was used to determine the inhibitory effect of the pre-treatment. Moreover, we subsequently focused on those main treatment drug concentrations causing the highest level of growth inhibition but not more than 75%. The presence of hysteresis was assessed by calculating the difference in AUC between the hysteresis and the corresponding control treatment with a two-sided Wilcoxon-rank-sum test adjusted for multiple comparison with FDR.

## Drug interactions

Drug interaction profiles were determined with the CombiANT approach[78] for a variety of antimicrobial combinations, and additionally a checkerboard assay for one specific combination, using both the for the wildtype PA14 and the CpxS T163P mutant. The CombiANT analysis is based on antibiotic diffusion in agar plates and has been previously validated against standard assays for assessing antibiotic interactions such as time kill and checkerboard assays[78–80]. Images of the CombiANT plate were taken with the BIO-RAD ChemiDoc Touch (Colorimetric blot, Epi white, medium size, auto optimal exposure), followed by image analysis with the CombiANT Imager v.3.5 and the nps_v3 analysis pipeline[81]. The interaction profile between strains was compared by two-sided Wilcoxon rank sum test for each antibiotic pair separately. A checkerboard approach was performed to assess the interaction between CAR and GEN. Seven concentrations of each antibiotic and three no-drug controls were distributed across a 96-well plate. The plates were incubated for 24 h at 37 °C with constant shaking and $OD_{600}$ measurements taken every 15 min, followed by inference of growth rates as described previously[82]. The degree of synergy was determined using the Bliss independence method[83] (Supplementary Fig. S9).

## Secondary structure analysis and 3D homology modeling of CpxS

Secondary structure analysis of the CpxS sequence was conducted using the Self-Optimized Prediction (SOPMA) program[84]. For constructing 3D homology modeling of CpxS, SWISS MODEL[85] server and HHpred server[86] were employed. The cytoplasmic portion (residues 175-445), comprising the HAMP, DHp, and CA domains, was modeled using the crystal structure of CpxA[62] (PDB ID: 4BIV) as the template. Concurrently, the homology model of the CpxS sensory domain (residues 8–148) was constructed by using the sensory domain from AbfS (sensor histidine kinase)[45] as the template (PDB ID: 2VA0). The composite full-length model of CpxS, excluding the transmembrane region (TM), was manually assembled in Coot[87]. Visualization of the final model was done using PyMol[88].

## Construction of mutants

Scar-free deletion mutants and specific *cpxS* mutants were generated using a two-step allelic exchange approach[89,90]. Briefly, fragments of approximately 700 bp up – and downstream of the mutation of interest were cloned into the plasmid pUIsacB[90] using Gibson

Assembly. The resulting plasmid was transformed into *E. coli* JM109. Tri-parental conjugation followed by two steps of homologous recombination on selective media was used to select for clones containing the mutation of interest. In addition, we also generated mutants with inducible *cpxS* overexpression in the wildtype PA14 and the *cpxS* deletion mutant, using a restriction digest – ligation set up. The *cpxS* open reading frame was cloned into the expression vector pME6032[91] and transferred to the two strains of interest. The expression of *cpxS* was induced by IPTG addition to the culture medium, followed by phenotypic analysis.

## Transcriptomics

The wildtype PA14 and the Cpx T163P mutant were subjected to a hysteresis or control treatment as above. After 30 min of pre-treatment, cells were collected by centrifugation and total RNA isolated, using the TRIzol lysis and the NucleoSpin miRNA kit (Macherey-Nagel), followed by rRNA depletion (RiboZero Bacterial Kit, Illumina), library preparation (TruSeq stranded total RNA, Illumina) and sequencing (Illumina, HiSeq4000, $1 \times 50$ bp). Read trimming was carried out via Trimmomatic[92], reads were followingly mapped and counted via Bowtie2[93] and EDGE-pro[94]. Normalization, differential expression analyses among treatments and strains as well as enrichment analyses were performed with *edgeR*[95], using the quasi-likelihood generalized negative binomial model and associated F tests to assess significant differential expression. We considered genes to be significantly differentially expressed if they showed the FDR-adjusted *p* value was below 0.05 and the absolute expression fold change was at least 2 (absolute $\log_2$ fold change $\geq 1$). Enrichment analyses were performed with clusterprofiler, using an FDR-adjusted *p*-value of 0.05 as cut-off. Reference genome as well as Gene Ontology (GO) and KEGG pathway annotation data were downloaded from the Pseudomonas Genome Database[96] (pseudomonas.com, June 2020).

## Membrane stress measurements

The impact of CAR pre-treatment on membrane stress was assessed by measuring general polarization with Laurdan for the wildtype PA14 and the CpxS T163P mutant, subjected to hysteresis and control treatments as above. After 15 min of pre-treatment, Laurdan was added at 100 μM (Supplementary Fig. S10). After a total of 30 min pre-treatment, the cells were collected by centrifugation, washed twice in M9 containing 0.08% DMF, a final resuspension in M9 and fluorescence measurements in an Infinite M Plex plate reader. The statistical significance of a difference between treatments was assessed with a two-sided Student's *t*-test.

## ELISA-based inference of intracellular GEN concentrations

Intracellular GEN concentration was measured for the wildtype PA14 and the Cpx T163P mutant subjected to hysteresis or control experiments as above ($n = 3$ per strain and treatment), followed by 3 x washing of cells in PBS, cell lysis (30 min at 37 °C in provided lysis buffer) and intracellular GEN ELISA measurements (Elabscience®, E-FS-E073), according to the manufacturer's instructions. The statistical significance of a difference between treatments was assessed with an ANOVA (Supplementary Table S11) followed by Tukey HSD (Honestly Significant Difference) test with FDR correction for multiple comparisons (Supplementary Table S12).

## Statistics & Reproducibility

All statistical analyses were conducted in R (version R 3.6.1 – 4.2.2). The detailed statistical tests for each experiment are given in the relevant Methods section. Further information on material and methods is provided in the supplement. The detailed results of the statistical analyses are provided in Supplementary Data files and the Supplementary Tables.

## Reporting summary

Further information on research design is available in the Nature Portfolio Reporting Summary linked to this article.

## Data availability

The source data of this study is provided in a supplementary Source Data file. The statistical data generated, as well as Key Resources used, are provided in the Supplementary Data files and as Supplementary Tables. The transcriptomic data generated in this study have been deposited in the NCBI's Gene Expression Omnibus under GEO Series accession number GSE290299. Source data are provided with this paper.

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

## Acknowledgements

We are very grateful to S. Hernando-Amado (Madrid, Spain), C. Pál (Szeged, Hungary), and T. Bollenbach (Cologne, Germany) for critical comments and advice on the manuscript. We further thank D. Rogers, J. Summers (Ploen, Germany) for guidance in allelic exchange, P. Rainey

(Ploen, Germany) for providing the plasmids and strains, then J. Lorenzen, K. Flinder, N. Steinbach, S. Butze (all Schulenburg lab), and L. Kirchhoff (Rupp lab) for supporting the experimental work, and also the Rupp and Schulenburg groups for general feedback. We are grateful for financial support from the German Research Foundation within the Research and Training Group 2501 (RTG 2501) on Translational Evolutionary Research (project 4.2 to H.S.), within the Excellence cluster Precision Medicine in chronic Inflammation (PMI; funding under Germany's Excellence Strategy EXC 2167-390884018, to B.K., K.R., J.R., H.S.), within the Clinician Scientist Program in Evolutionary Medicine (CSEM) – project number 413490537 (to EEG), and as part of the individual grants SCHU 1415/12-2 (to H.S.) and BR-2915/7-1 (to M.B.). We are grateful for financial support from the Swedish Research Council, project number 2021-02091 (to D.I.A.). We are also grateful for financial support from the Max-Planck Society (Fellowship to H.S.), the Leibniz Association within the Leibniz Science-Campus Evolutionary Medicine of the Lung (EvoLUNG, to H.S.), and the project SKILLED funded by the DAMP foundation (to J.R., H.S.). This work was also supported by the ZMB Young Scientist award and the FWF grant 10.55776/ESP219 (to R.R.) and the TransEvo Innovation prize (to F.B.). The funders had no role in study design, data collection and interpretation, or the decision to submit the work for publication.

## Author contributions

Conceptualization: F.B., L.U., L.T., E.E.G., J.R., D.I.A., H.So, M.B., R.R., H.Schu. Formal analysis: F.B., L.U., L.T., E.E.G., D.S., A.S.S., R.R.; Funding acquisition: F.B., K.F.R., J.R., R.R., H.Schu. Investigation and methodology: F.B., L.U., L.T., E.E.G., K.H., D.S., A.S.S., A.B., S.P., S.B., D.N., B.N.D., S.F., B.K., R.R.; Resources: E.E.G., B.K., K.F.R., J.R., H.Schu. Supervision: F.B., L.U., L.T., E.E.G., D.N., J.R., D.I.A., H.So, M.B., R.R., H.Schu. Writing of original draft: F.B., L.U., R.R., H.Schu. Review and editing: all authors.

## Funding

## Competing interests

The authors declare the following competing interest: Access to the CombiANT technology was provided by Rx Dynamics AB as a product BETA test. R.R. and D.I.A. are co-founders of Rx Dynamics AB. The company had no influence on study design, investigation, data analysis, manuscript writing and decision to publish. Apart from this, the authors declare no conflict of interest.

## Additional information

[1]Department of Evolutionary Ecology and Genetics, University of Kiel, Kiel, Germany. [2]Institute of Medical Microbiology, University Hospital Schleswig-Holstein, Lübeck, Germany. [3]LungenClinic Grosshansdorf, Großhansdorf, Germany. [4]Airway Research Center North (ARCN), Member of the German Center for Lung Research (DZL), Großhansdorf, Germany. [5]Department of Medicine, University of Kiel, Kiel, Germany. [6]Institute for General Microbiology, University of Kiel, Kiel, Germany. [7]CSSB Centre for Structural Systems Biology, Deutsches Elektronen-Synchrotron DESY, Hamburg, Germany. [8]Department of Biosciences and Bioengineering, Indian Institute of Technology Dharwad, Dharwad, India. [9]Competence Centre for Genomic Analysis Kiel, University of Kiel, Kiel, Germany. [10]Clinical Infectious Disease, Research Center Borstel, Leibniz Lung Center, Borstel, Germany. [11]German Center for Infection Research (DZIF), Hamburg-Lübeck-Borstel-Riems, Germany. [12]Respiratory Medicine & International Health, University of Lübeck, Lübeck, Germany. [13]Infectious Diseases Clinic, University Hospital Schleswig-Holstein, Lübeck, Germany. [14]Department of Medical Biochemistry and Microbiology, Uppsala University, Uppsala, Sweden. [15]Institute of Science and Technology Austria, Klosterneuburg, Austria. [16]Max Planck Institute for Evolutionary Biology, Ploen, Germany. [17]These authors contributed equally: Florian Buchholz, Lina M. Upterworth. [18]These authors jointly supervised this work: Roderich Roemhild, Hinrich Schulenburg. ✉e-mail: roderich.roemhild@ist.ac.at; hschulenburg@zoologie.uni-kiel.de

