## [Transparent Peer Review file · Nature Communications]

Robust antibiotic sensitization of pathogenic *Pseudomonas aeruginosa* via negative hysteresis in the cell envelope

Corresponding Author: Professor Hinrich Schulenburg

Version 0:

Reviewer comments:

Reviewer #1

(Remarks to the Author)

This is a well-executed study that investigates negative hysteresis (NH), an important biological process with implications for treating multidrug-resistant infections. This research builds on a previous finding of NH in the *Pseudomonas aeruginosa* laboratory strain PA14, where β -lactam treatment increased sensitivity to aminoglycosides (Roemhild et al. 2018). The original experiment identified a SNP in *cpxS* (T163P) that abrogated NH, indicating a candidate molecular mechanism.

The *Cpx* operon is well-characterized in *Escherichia coli* as a regulator of efflux pump expression (Raivio and Silhavy 2001), but its ortholog in *P. aeruginosa* is structurally distinct and less understood. Previous studies in *P. aeruginosa* elucidated the function of *cpxS* as a sensor kinase that works in tandem with the *cpxR* response regulator to influence expression of the *mexAB-oprM* efflux pump (Tian et al. 2016; Tian and Wang 2023). Mutations in *cpxS* have also been identified in *P. aeruginosa* isolated from patients (Hernando-Amado et al. 2025), underscoring clinical relevance.

This study extends existing knowledge of NH and the *Cpx* operon in *P. aeruginosa* by including multiple antibiotics, performing tests in laboratory strains and clinical isolates, and investigating the physiological impacts and molecular basis of NH. Their key findings are that pretreatment with three β -lactams (meropenem, piperacillin/tazobactam, and ceftazidime) induces NH with gentamicin, that this occurs regardless of genetic background (across all strains of *P. aeruginosa* tested), and that this can occur even if the strain is already resistant to gentamicin. They also propose a model for the *Cpx* operon in *P. aeruginosa* that functions similarly to *Cellvibrio japonicus*; this was verified by demonstration that membrane stress reduces permeability by aminoglycosides, that *cpxS* is strongly upregulated during the membrane stress response, and that *cpxS* and *cpxP* operate in a negative feedback loop.

Major comments

1. In general there isn't much review of the literature to contextualize results. Also lacking is a discussion of what these findings indicate for clinical treatment.
2. How common is hysteresis? Only a few examples are cited (refs 12, 19, and 23), suggesting it is either uncommon or poorly studied. Please add an explanation of why it is uncommon or understudied and what the current outstanding questions are. Related to this, on line 174: please indicate out of how many trials the 121 examples of hysteresis were found.
3. The T163P mutation in *cpxS* is a well-studied mutation, and the Introduction/Discussion of this paper lacks a review of what is known about and has already been done with this mutation.
4. The assay designed to address chronic infections by using mixed populations (Line 193-202) is not explained in sufficient detail within the main text. Please indicate in the text how many populations you constructed and tested from each patient. Additionally:
 - a. It appears that only *P. aeruginosa* were isolated from the patient samples (there were likely other bacteria present, so this is a weakness of the approach that should be acknowledged).
 - b. Did the isolates exhibit chronic phenotypes like drug resistance or mucoidy?
 - c. Line 768 (legend for Fig. 3D) what are the time points and why did you only consider multiple time points for a few populations?

Minor comments

1. Fig 2A: the CIP results are mentioned only in the figure caption, please describe them in the main text.

2. The reference to “standard hysteresis treatment” (Line 295), was confusing since you previously tried inducing hysteresis with multiple drugs. Perhaps rephrase to say this analysis was done with and without pre-treatment with CAR.
3. In your supplemental methods, please indicate why you used 3 methods to measure MIC and whether you calibrated across methods to ensure comparability.
4. Fig. S2 has incongruent colors: e.g., the legend indicates blue for CAR pretreatment but there is no blue in the figure, and there’s a black line in the figure but no black line in the legend.

Conclusion

NH could be a broadly significant concept that enhances our understanding of antibiotic resistance and can be used to address the urgent challenge of treating multidrug-resistant infections. This study demonstrates that temporally fluctuating antibiotic regimes may offer an effective treatment strategy that does not require new drug discovery). As an ESKAPE pathogen, *P. aeruginosa* is an appropriate and relevant model system.

Reviewer #2

(Remarks to the Author)

Review of Bucholz et al. Robust antibiotic sensitization of pathogenic *Pseudomonas aeruginosa* via negative hysteresis in the cell envelope

In this article, the authors test for hysteresis (when pre-treatment to one antibiotic influences subsequent resistance to other antibiotics) in *P. aeruginosa*. Our overall assessment is that the authors do not present convincing evidence of hysteresis, and the manuscript has important flaws, including non-replicated experiments and incomplete statistical methodology. The authors present convincing evidence that the cpx pathway is involved in antibiotic sensitization. However, our overall assessment is that this manuscript is not suitable for publication, and we recommend rejection.

Major points

1. Methods used to test for hysteresis. To test for hysteresis the authors pre-treat bacteria with one antibiotic and then exchange the media to another antibiotic. The problem with this method is that the antibiotic used for pre-treatment enters either the cytoplasm of bacterial cells (aminoglycosides ie GEN, fluoroquinolones ie CIP) or the periplasmic space (ie B-lactam antibiotics CAR, MER, PIT, CTZ). Exchanging the media creates a diffusion gradient that will favour the diffusion of the pre-treatment antibiotic out of the cell, but this diffusion will be far from instantaneous, especially in *Pseudomonas* which is notorious for having cells with low permeability. This creates a fundamental problem, as the authors cannot discriminate between effects or pre-treatment that are caused by the damage caused to cells and/or physiological responses to the antibiotics or if the pre-treatment effects are simply the result of the persistence of the pre-treatment antibiotics in bacterial cells when the ‘main’ antibiotic is administered.

In Figure 2, the authors show that negative hysteresis is most common among pairs of antibiotics that show synergistic interactions. This key result is consistent with the idea that the negative hysteresis effect could simply be due to the interaction between retained pre-treatment antibiotic and the second antibiotic used for treatment. The short duration of the hysteresis effect (Figure 1F) is also consistent with this idea, and may reflect the time needed for pre-treatment antibiotics to diffuse out of cells.

2. Experimental design and experimental method. The ‘gold standard’ method to measure the impact of antibiotics on bacteria viability is to measure how antibiotic exposure influences live bacteria cells counts. The authors use this approach to test for interactions between GEN and CAR in Figure 1. However, the experiments reported in this Figure are not replicated, making it impossible to assess the validity of these results.

The authors then go on to test for hysteresis between pairs of antibiotics in clinical isolates. The authors based their measurements of interactions using OD curves. We appreciate that assays like this require high throughput experimental methods, but there are a number of important potential biases and limitations with using OD to measure responses to antibiotics. For example, exposure to B-lactam and fluoroquinolone antibiotics causes changes to bacterial cell morphology (ie cell filamentation) that alter the correlation between cell number and OD. Antibiotics also cause cell death, and OD does not allow for live and dead cells to be distinguished. Owing to these potential limitations, using OD to measure the impact of antibiotics on bacterial growth requires robust validations using methods that assess viable cell counts. We are not convinced by the validation provided by the authors, and we suggest that they should randomly test a fraction of the interactions in their screen instead of cherry picking 1 or 2 interactions, particularly if these are previously known examples of hysteresis.

3. The use of clinical isolates to test drug interactions is a positive aspect of this study, particularly given that drug interactions and collateral sensitivity are often poorly conserved across strains. However, only a subset of clinical isolates displayed ‘robust’ negative hysteresis, with only two COPD populations, many switches were neutral (lines 197-198). The manuscript would benefit from a clearer framework for why some isolates or resistance phenotypes fail to show the phenomenon.

4. The authors identify the well-known Cpx envelope stress pathway as a key regulatory mechanism. Combining targeted gene deletions, overexpression constructs, and transcriptomic profiling provides a good methodological basis for mechanistic insights into how envelope stress leads to antibiotic sensitisation. The authors provide convincing evidence that the cpx pathway is involved in antibiotic sensitization – however, the data for cpxS mutants (e.g., T163P, G188S) at times yield unexpected outcomes (lines 235-237). Some mutants do not replicate the wild-type phenotype or lead to positive hysteresis. Clarification is needed on how these differences influence the interpretation that CpxS is a key driver of negative hysteresis. Additional biochemical or phenotypic assays (e.g., membrane permeabilisation assays) might help to confirm how each mutation modifies envelope stress responses.

5. A further limitation of this study is that the statistical methodology is not described very clearly, making it difficult to assess the methods used to analyze the data. For example, it is unclear if the authors treated different measures of the same culture at different time points as independent observations or not (they should not be considered independent) and the degree to which their results were based on comparisons between biological and technical replicates.

Overall assessment: Our overall assessment is that we are not convinced that hysteresis is a real phenomenon or if the authors have effectively just developed a new assay to measure drug-drug interactions (point 1). This is a fundamental problem with the study. The authors are forthcoming about how drug pharmacokinetics may make it difficult to apply sequential antibiotic treatment in the context of infections, but the strong dose dependency (ie comparing results with 1mg/L vs. 2 mg/L GEN in Figure 1B) and transient nature of these interactions (ie Figure 1F) also make us skeptical about the potential applications of this work for treating bacterial infections.

Reviewer #3

(Remarks to the Author)

Reviewer #4

(Remarks to the Author)

Version 1:

Reviewer comments:

Reviewer #1

(Remarks to the Author)

My prior review was largely supportive and requested more clarity about methods and some greater context from the literature, which the authors addressed well.

However Reviewer 2 raised important methodological concerns that drew my interest. The biggest one is that the methods didn't preclude carryover of pretreatment antibiotic in the periplasm/cytoplasm of cells, so you can't distinguish whether NH is a real, post-drug effect, or an effect of residual intracellular drug. Reviewer 2 also points to the transience of NH as consistent with drug carryover, as the effect could be disappearing once the residual drug diffuses out of cells.

The authors did a lot of additional work to address this, but not a direct test (I think you'd need mass spectrometry to estimate intracellular drug concentration). While not totally satisfying, it convinced me that NH is real even if its underlying biology remains unclear. Specific points about this follow:

- a. The authors did additional experiments where they wash pretreated cells 2x before their assay for NH. However, I don't think you can wash away intracellular drug.
- b. They also added experiments showing that other membrane stresses can cause NH: 1) mild heat shock and 2) CCCP (depletion of membrane potential). From this they conclude that "abrupt changes to the membrane fluidity can trigger hysteresis". This proves their principle, but it's not a direct test.
- c. They comment: "If antibiotic retention were the sole explanation, we would expect enhanced killing regardless of the second antibiotic used". It is unclear to me why all drug combinations should produce the same outcome -- wouldn't this depend on mechanism of action?
- d. Finally, the authors argue that increased efflux would expel residual drug and abolish NH. They found that a mexR mutation increased resistance but had no effect on NH, suggesting there NH wasn't caused by intracellular drug. I believe this because they confirmed that the mutation caused increased efflux via resistance. If increased efflux doesn't affect NH, then NH can't all be due to residual drug.
- e. The authors don't clearly address that the transience of the hysteresis phenotypes may also be a corollary of residual drug diffusion. Rather, they say that NH is not the same as drug interactions because they get different NH types depending on the order of the drug, but it does not mean they are not related.

Reviewer #2

(Remarks to the Author)

The authors have made substantial changes to the content and text of the manuscript following peer review. Most of our concerns have been addressed, and the authors have substantially improved this manuscript, apart from one technical

concern listed below. Addressing this concern will not substantially alter the conclusions of this manuscript. The only bigger issue that we have with the paper is the link between this work and the authors' previous paper on hysteresis (<https://www.pnas.org/doi/epdf/10.1073/pnas.1810004115>). It is clear that this paper has advanced our understanding of hysteresis, but we found ourselves in a position where we were questioning whether the advances in understanding presented in this paper were sufficient to warrant publication in a journal like Nat Comms. We think that this is a debatable point, and the editor may wish to consider this issue if it is raised as an issue by other reviewers.

Outstanding issue:

The authors have carried out paired measures of OD and CFUs to test the validity of their approach (Figure S5). When gentamicin is used as the main antibiotic for treatment the OD values and CFUs are well correlated, validating the use of OD. However, the correlation between microtiter plate OD and CFUs is not convincing in any way when B-lactams are used as the antibiotic for main treatment. The authors test for a correlation between logOD and logCFU by linear regression, and this analysis is flawed. The underlying data do not show a bivariate normal distribution (ie normally distributed x and y values) violating a central assumption of linear regression. Violating this assumption tends to lead to inflated r^2 values (for example when linear regression is on data sets that have two 'clouds' of points) and we argue that their estimated (low) r^2 value of 0.27 is an overestimate of the true strength of association – cultures with OD of 0.05 and .5 were associated with similar viable cell densities. The authors should drop all of the data from plate reader based assays where B-lactams were used as the main treatment.

Reviewer #3

(Remarks to the Author)

Reviewer #4

(Remarks to the Author)

Version 2:

Reviewer comments:

Reviewer #2

(Remarks to the Author)

The authors have responded to my technical concerns, but I remain unconvinced by their arguments regarding the novelty and importance of this work, as outlined below and quoting from their response to referees.

(i) it provides the first evidence of the widespread and robust expression of negative hysteresis across genomic backgrounds for a particular pathogen taxon,

Demonstrating negative hysteresis in clinical isolates is an advance with respect to the authors' previously published work. However, I consider this to be an incremental advance as it is effectively confirming a previously reported finding. I think that the authors case here would have been much stronger if they had been able to uncover why clinical isolates differ in their antibiotic responses.

(ii) it provides the first evidence of the robust expression of negative hysteresis upon switches from a beta lactam drug to an aminoglycoside which is beyond the previously reported switch from carbenicillin to gentamicin,

I view this as being quite a minor point. Given the similarities in structure and mechanisms of action of different antibiotics from the same class it would be surprising if this negative hysteresis between B-lactams and aminoglycosides was not widespread. It would have been much more novel if, for example, the authors had been able to demonstrate negative hysteresis across a wide range of combinations of antibiotic class.

(iii) it provides evidence that negative hysteresis can be induced in resistant strains and pathogen populations isolated from patients, thereby highlighting novel, currently not considered treatment options,

I think that demonstrating the applicability of negative hysteresis to clinical isolates is important, but the authors present very little data on this in the manuscript – the data in Figure 3 is from isolates collected from n=3 patients!

(iv) it provides novel information on the molecular basis of negative hysteresis that relies on beta-lactam-induced changes in membrane fluidity and increased intra-cellular concentration of the subsequently administered aminoglycoside drug, and (v) it provides new evidence for the central role of the Cpx cell envelope stress response system in mediating hysteresis based on our detailed functional genetic analysis of mutant strains previously not available (e.g., loss-of-function, gain-of-function, overexpression mutants, as well as mutants of related genes).

I agree with the authors that they have substantial mechanistic detail to their previously published case of negative hysteresis. The mechanistic work reported by the authors provides a sufficient level of analysis as supporting work, but it does not rise to the level that I would expect, for example, from a paper on AMR mechanisms. Providing supporting

mechanistic detail into a previously published phenomenon does not, in my view, constitute the kind of novelty that I expect from papers in Nature Communications.

Reviewer #3

(Remarks to the Author)

Dear editors,

We are pleased to submit our revised manuscript to *Nature Communications*. We carefully considered the reviewers comments to improve our manuscript. Importantly, we performed several new experiments for a much more comprehensive demonstration of the cellular and physiological changes underlying the phenomenon of negative hysteresis. The results of these new experiments as well as a refined analysis of our data have now been included in the revised manuscript and they clearly strengthen our key conclusions. See Figures 1B, 1C, 1D, 1E, 5C, 5D, S2A, S2B, and S5. Moreover, we further carefully refined our arguments and the interpretation of results, in response to the helpful advice given by the reviewers. Therefore, we hope that our manuscript is now acceptable for publication.

Negative hysteresis is a unique and understudied phenomenon that we show for the first time to be common and robustly expressed in the high priority human pathogen *Pseudomonas aeruginosa*, that we show for the first time to be caused by cell envelope stress, involving the two-component Cpx system, and that we identify as a promising focus for optimizing antibiotic therapy.

Please let us know if you have any questions!

Yours sincerely,

Roderich Roemhild, Hinrich Schulenburg (on behalf of all authors)

Response to reviewer comments

Below, the original reviewer comments are given in *gray and italics*, while our response is given in **blue**. We further provide manuscript and supplement files, in which changes are tracked and in which we use **comments to indicate where we modified the manuscript in response to reviewer comments**.

Dear Professor Schulenburg,

I am writing from Nature Communications on behalf of Dr Rita Ramalhete.

Thank you for submitting your manuscript, 'Robust antibiotic sensitization of pathogenic Pseudomonas aeruginosa via negative hysteresis in the cell envelope'. It has now been seen by four referees whose comments are provided below this message.

For your information, I have provided the expertise of the referees immediately below.

Referees 1 and 4: Antimicrobial resistance evolution, microbial genetics and evolution

Referees 2 and 3: Pseudomonas aeruginosa, evolution and genomics of antimicrobial resistance, hysteresis

As you will see from the reports, while Referees 1 and 4 find the work of potential interest, Referees 2 and 3 have raised serious concerns about the strength of support for the central conclusions relating to hysteresis. In particular, we share their concerns about the lack of replication, the potential bias in the hysteresis screening based on the potential impact of antibiotic usage on cell wall permeability, and the lack of validation of the OD curves including a lack of random testing of drug-drug interactions. In light of these concerns, I am afraid that we will not offer to publish the manuscript.

I am sorry that we cannot be more positive on this occasion and thank you for the opportunity to consider your work.

Best regards,

Ripu

*Ripudaman K Bains, PhD
Chief Editor for Microbiology and Infectious Diseases
Nature Communications*

Reviewers' comments:

Reviewer #1 (Remarks to the Author):

*This is a well-executed study that investigates negative hysteresis (NH), an important biological process with implications for treating multidrug-resistant infections. This research builds on a previous finding of NH in the *Pseudomonas aeruginosa* laboratory strain PA14, where β -lactam treatment increased sensitivity to aminoglycosides (Roemhild et al. 2018). The original experiment identified a SNP in *cpxS* (T163P) that abrogated NH, indicating a candidate molecular mechanism.*

*The *Cpx* operon is well-characterized in *Escherichia coli* as a regulator of efflux pump expression (Raivio and Silhavy 2001), but its ortholog in *P. aeruginosa* is structurally distinct and less understood. Previous studies in *P. aeruginosa* elucidated the function of *cpxS* as a sensor kinase that works in tandem with the *cpxR* response regulator to influence expression of the *mexAB-oprM* efflux pump (Tian et al. 2016; Tian and Wang 2023). Mutations in *cpxS* have also been identified in *P. aeruginosa* isolated from patients (Hernando-Amado et al. 2025), underscoring clinical relevance.*

*This study extends existing knowledge of NH and the *Cpx* operon in *P. aeruginosa* by including multiple antibiotics, performing tests in laboratory strains and clinical isolates, and investigating the physiological impacts and molecular basis of NH. Their key findings are that pretreatment with three β -lactams (meropenem, piperacillin/tazobactam, and ceftazidime) induces NH with gentamicin, that this occurs regardless of genetic background (across all strains of *P. aeruginosa* tested), and that this can occur even*

if the strain is already resistant to gentamicin. They also propose a model for the Cpx operon in P. aeruginosa that functions similarly to Cellvibrio japonicus; this was verified by demonstration that membrane stress reduces permeability by aminoglycosides, that cpxS is strongly upregulated during the membrane stress response, and that cpxS and cpxP operate in a negative feedback loop.

Major comments

1. In general there isn't much review of the literature to contextualize results. Also lacking is a discussion of what these findings indicate for clinical treatment.

Response: We appreciate the reviewer's suggestion. We have now expanded the Introduction and provide a more detailed description of the characteristics of hysteresis, the timeline of its discovery, the importance of the directionality of effects, and also its relationship with drug interaction type as well as evolved collateral sensitivity, always including citation of relevant publications.

Moreover, in the revised discussion, we discuss the relevance of our findings for clinical treatment. We note that the robustness of negative hysteresis upon beta-lactam – aminoglycoside switches across the genomic diversity of *P. aeruginosa* make this phenomenon a highly promising focus for optimizing treatment designs, especially because it likely “eliminates the need for specific prior phenotypic tests”. Moreover, we point out that the observation of negative hysteresis in strains resistant to the pre-treatment antibiotic (most effectively a beta-lactam antibiotic) may “provide the unique opportunity for re-introducing β -lactams into treatment regimens that otherwise may have been deemed ineffective”. We further highlight that negative hysteresis likely “underlies the previously published increase in *P. aeruginosa* eradication and health improvements in a cohort of infected CF patients, who were subjected to a sequential treatment of first a β -lactam followed by an aminoglycoside drug”, thereby further indicating its high promise as a focus for optimizing antibiotic therapy. Finally, we note that “Cpx system of *P. aeruginosa* also regulates virulence”, indicating “a clinical relevance of genetic variation in cpxS” and “further highlighting the need for a better understanding of this stress response system”.

2. How common is hysteresis? Only a few examples are cited (refs 12, 19, and 23), suggesting it is either uncommon or poorly studied. Please add an explanation of why it is uncommon or understudied and what the current outstanding questions are. Related to this, on line 174: please indicate out of how many trials the 121 examples of hysteresis were found.

Response: There are indeed only very few studies that examined the phenomenon of hysteresis and this is exactly the point of our manuscript: to draw attention to this fascinating phenomenon and stimulate further research on antibiotic-inducible effects on susceptibility. This is a phenomenon that is understudied and we currently lack basic information on its distribution and the underlying mechanisms. Our study addresses both of these points, using the human opportunistic pathogen *Pseudomonas aeruginosa* as a model. These aspects are now more clearly highlighted in the revised introduction as well as discussion.

Regarding the last point: We tested 412 antibiotic switches and identified 121 significant hysteresis cases across our panel, where most are related to a beta-lactam - gentamicin drug switch. This information was added to the revised Results.

3. The T163P mutation in cpxS is a well-studied mutation, and the Introduction/Discussion of this paper lacks a review of what is known about and has already been done with this mutation.

Response: Many thanks for this comment. We are happy to provide a more detailed overview of the role of *cpxS* and the T163P mutation in antimicrobial resistance in our revised manuscript.

We are aware of three additional studies, which performed a functional genetic analysis of the CpxS T163P mutation, including its reconstruction in a defined genetic background, followed by its phenotypic characterization, including the articles of Galdino et al. 2024 mBio (<https://doi.org/10.1038/s41564-024-01601-4>), Chan et al. 2023 ACS Infect Dis (<https://doi.org/10.1021/acsinfecdis.3c00167>), and our own previous work (Roemhild et al. 2018 PNAS, doi:10.1073/pnas.1810004115), the latter of which however only included a very baseline characterization of this mutation.

We are aware of further four studies that identified the T163P mutation in antibiotic resistant *P. aeruginosa* genomes, including Gomis-Font et al. 2023 JAC (<https://doi.org/10.1093/jac/dkad172>), Hernando-Amado et al. 2025 Nat Comm (<https://doi.org/10.1038/s41467-025-58597-6>) and again 2 of our own papers (Barbosa et al. 2017, 2021; <https://doi.org/10.1093/molbev/msx158>; <https://doi.org/10.1093/molbev/msaa233>).

Six more studies linked mutations in *cpxS* (although not specifically T163P) to changes in antibiotic susceptibility (<https://doi.org/10.1371/journal.ppat.1005932>; <https://doi.org/10.1128/aac.00921-23>; <https://doi.org/10.1371/journal.pbio.2001586>; <https://doi.org/10.1128/aac.01009-23>; <https://doi.org/10.1128/mBio.00517-17>; <https://doi.org/10.1098/rsob.150175>).

We agree with the reviewer that the available studies on the T163P mutation provide valuable context for our results. Therefore, we now discuss these previous studies in the context of our new findings in two paragraphs of our revised Discussion.

4. The assay designed to address chronic infections by using mixed populations (Line 193-202) is not explained in sufficient detail within the main text. Please indicate in the text how many populations you constructed and tested from each patient.

Response: We are happy to provide more information on the analysed populations isolated from patients. We reconstructed *P. aeruginosa* populations with isolates obtained from 4 COPD patients. For two of these patients (patients 3 and 4), we had isolates from two time points during treatment and therefore, for each of these patients, we reconstructed two populations. The considered populations

consisted of at least 8 up to 14 isolates. We now provide this additional information in the revised Results section, the revised Methods and the extended Methods in the Supplement. A detailed list of how many individual isolates were included is also available within the SI – Key Resource Table

Additionally:

a. It appears that only P. aeruginosa were isolated from the patient samples (there were likely other bacteria present, so this is a weakness of the approach that should be acknowledged).

Response: For the reconstruction of populations, we specifically and on purpose focused on *P. aeruginosa*, which serves as the model pathogen for our current study. For your information: Additional bacterial species were isolated in 3 of the patient samples, and in these cases, only single isolates of the other species; the 3 remaining patient samples only contained *P. aeruginosa*. None of the considered patient samples included *S. aureus*. We now provide this additional information in the revised Methods, the revised extended Methods in the Supplement, and also in the Supplementary Table S21.

b. Did the isolates exhibit chronic phenotypes like drug resistance or mucoidy?

Response: The considered patient populations varied in resistance towards antibiotics. None showed a mucoid phenotype. Baseline information on the characteristics such as morphology and MIC values of these patient isolates are now provided in a new Supplementary Table (Supplementary Table S22).

c. Line 768 (legend for Fig. 3D) what are the time points and why did you only consider multiple time points for a few populations?

Response: We only considered two time points for two of the patients, because only these two patients yielded pathogen-containing cultures after the initial treatment. We added more information on the patient populations as already described at the beginning of our response to this major comment 4.

Minor comments

1. Fig 2A: the CIP results are mentioned only in the figure caption, please describe them in the main text.

Response: This has now been addressed in the revised Results section.

2. The reference to “standard hysteresis treatment” (Line 295), was confusing since you previously tried inducing hysteresis with multiple drugs. Perhaps rephrase to say this analysis was done with and without pre-treatment with CAR.

Response: We followed the reviewer’s suggestion and refined the text accordingly at the end of the revised Results section.

3. In your supplemental methods, please indicate why you used 3 methods to measure MIC and whether you calibrated across methods to ensure comparability.

Response: We here used 3 different methods to measure MIC, depending on how precise the MICs had to be for the following assays. The time-kill assays need the most precise MICs to hit the desired inhibitory concentration; therefore, MIC were determined using broth micro-dilution (BMD). For the large screen, we used either the Vitek2 approach or E-tests to obtain reference MICs and then tested 2 different pre-treatment concentrations and 4 different main treatment concentrations. Within the screen, we always selected the treatment that induced the highest amount of inhibition (up to 75%) for our analysis. Importantly, we used only one set of MIC determination method per assay type or panel, including respective controls, thereby ensuring consistency and comparability of results per assay/panel. MIC-values inferred from either the BMD approach or the Vitek/E-test approach were never compared or used as a basis for the same assay, thus not requiring any kind of cross-calibration. These points are now explained in the revised Methods and the revised Supplementary Methods.

4. Fig. S2 has incongruent colors: e.g., the legend indicates blue for CAR pretreatment but there is no blue in the figure, and there's a black line in the figure but no black line in the legend.

Response: We now corrected the colors and legend text of the original Fig. S2. We have now moved this Figure to the main text as Fig. 1E.

Conclusion

NH could be a broadly significant concept that enhances our understanding of antibiotic resistance and can be used to address the urgent challenge of treating multidrug-resistant infections. This study demonstrates that temporally fluctuating antibiotic regimes may offer an effective treatment strategy that does not require new drug discovery). As an ESKAPE pathogen, P. aeruginosa is an appropriate and relevant model system.

Response: We thank the reviewer for their overall positive evaluation of our study and the constructive comments on our manuscript.

Reviewer #2 (Remarks to the Author):

Review of Bucholz et al. Robust antibiotic sensitization of pathogenic Pseudomonas aeruginosa via negative hysteresis in the cell envelope

In this article, the authors test for hysteresis (when pretreatment to one antibiotic influences subsequent resistance to other antibiotics) in P.aeruginosa. Our overall assessment is that the authors do not present convincing evidence of hysteresis, and the manuscript has important flaws, including non-replicated experiments and incomplete statistical methodology. The authors present convincing evidence that the cpx pathway is involved in antibiotic sensitization. However, our overall assessment is that this

manuscript is not suitable for publication, and we recommend rejection.

Major points

1. Methods used to test for hysteresis. To test for hysteresis the authors pre-treat bacteria with one antibiotic and then exchange the media to another antibiotic. The problem with this method is that the antibiotic used for pre-treatment enters either the cytoplasm of bacterial cells (aminoglycosides ie GEN, fluoroquinolones ie CIP) or the periplasmic space (ie B-lactam antibiotics CAR, MER, PIT, CTZ). Exchanging the media creates a diffusion gradient that will favour the diffusion of the pre-treatment antibiotic out of the cell, but this diffusion will be far from instantaneous, especially in Pseudomonas which is notorious for having cells with low permeability. This creates a fundamental problem, as the authors cannot discriminate between effects or pre-treatment that are caused by the damage caused to cells and/or physiological responses to the antibiotics or if the pre-treatment effects are simply the result of the persistence of the pre-treatment antibiotics in bacterial cells when the 'main' antibiotic is administered.

Response: We thank the reviewer for raising this important point. In response, we now revised the manuscript and included an additional experiment that tested for the role of antibiotic retention by comparing the presence of hysteresis without washing and with 2 PBS washes. In both cases we observed significant negative hysteresis showing that negative hysteresis is not explained by retention or at least retention alone. We added a new Supplementary Figure (Fig. S2B) and point to the new data in the revised Results section. We further emphasize in the revised Results section that covalently bound CAR and/or any CAR-induced changes in the cellular physiology are likely to be sufficient for hysteresis. We additionally discuss the possibility of antibiotic retention in the revised Discussion.

Additionally, our data clearly demonstrate that negative hysteresis arises from physiological and cellular changes induced by the initial antibiotic, which modulate the bacterial response to the second antibiotic. Specifically:

- Figure 5 of our original and also revised manuscript does unequivocally demonstrate that the pre-treatment antibiotic does cause a cellular and physiological change (Fig. 5A, 5B) that then causes increased uptake of the second antibiotic GEN (now Fig. 5E).
- We have now added the results of two newly performed experiments that further strengthen the link between CAR induced cellular and physiological changes and negative hysteresis. Firstly, we tested whether the observed changes in membrane fluidity are responsible for negative hysteresis. An orthogonal way to modify membrane fluidity are sudden shifts in temperature, to which bacteria respond by adjusting the membrane fluidity. We therefore tested whether a mild heat shock by a sudden shift from 37°C to 50°C for 15 min, ahead of the pre-treatment step can induce negative hysteresis. Consistent with the membrane stress hypothesis, we expected (i) the heat shock by itself to accelerate GEN killing and (ii) the heat shock to suppress the possibly redundant inducing effect of CAR. The experiment confirmed both predictions (Fig. 5C, statistical results in Supplementary Table S17), supporting our model that abrupt changes to the membrane fluidity trigger the CAR induced GEN hysteresis. See additions to the Results section, the Discussion, the Methods, and also new Fig. 5C.
- Secondly, the uptake of aminoglycoside is energized by the proton motive force of the cytoplasmic membrane. Thus, induced dynamics of proton motive force may link pre-treatment inhibition to

hysteresis lethality. The chemical inhibitor CCCP allows protons to cross the cytoplasmic membrane, collapsing membrane potential such that no dynamic change of the membrane is possible. When we pre-incubated cells with CCCP, cell numbers did not increase, relative to controls treated with solvent only. This reduction is likely explained by a requirement of membrane potential for growth. The pre-treatment with only CAR was more effective than CCCP at inducing negative hysteresis. Intriguingly, the CCCP incubation precluded any additional effects of CAR pre-treatment for hysteresis (New Fig. 5D, statistical results in new Supplementary Table S18), further supporting the idea that an altered cell envelope underpins negative hysteresis. See additions to the Results section, the revised Discussion and Methods, as well as the new figure Fig. 5D.

- Our genetic analysis of *cpxS* provides further evidence for a physiological change underlying hysteresis. The gain-of-function variant CpxS T163P does not only abolish the negative-hysteresis phenotype (see Figs. 4B, 4C, 4D), it does so by abrogating the effects of the pre-treatment drug on envelope stress, membrane stress, and also uptake of the second antibiotic GEN. These findings are shown by the results in Figs. 5A, 5B, and 5E.
- If antibiotic retention were the sole explanation, we would expect enhanced killing regardless of the second antibiotic used. However, our hysteresis matrix (Fig. 2A) demonstrates that enhanced killing is limited to specific antibiotic switches. In several cases, switching antibiotics even confers a protective effect.
- The idea that the retained pre-treatment antibiotic causes enhanced killing of a subsequently administered antibiotic is related to the phenomenon of the post-antibiotic effect. We specifically tested this and we did not find any evidence for such a post-antibiotic effect, as illustrated in Supplementary Figure S2A. Here, exposure to only the pre-treatment antibiotic does not cause any bacterial death, but rather normal growth, as in the no-drug control.
- If retained residues of the first antibiotic is key, then increased expression of efflux pumps mediating export of such drugs should abolish the observed killing effect. In our previous work, we tested a *mexR* mutant (MexR is a negative regulator of the MexAB-OprM efflux pump) that should lead to enhanced export of beta-lactams like our pre-treatment antibiotic Carbenicillin, and did indeed observe an increase in drug resistance. Importantly, we did not find that this *mexR* mutation has any effect on the hysteresis phenotype (see Figs. 5A and 5C in Roemhild et al. 2018 PNAS, <https://www.pnas.org/cgi/doi/10.1073/pnas.1810004115>).

We now integrated these points in our revised Results and our revised discussion.

In Figure 2, the authors show that negative hysteresis is most common among pairs of antibiotics that show synergistic interactions. This key result is consistent with the idea that the negative hysteresis effect could simply be due to the interaction between retained pretreatment antibiotic and the second antibiotic used for treatment. The short duration of the hysteresis effect (Figure 1F) is also consistent with this idea, and may reflect the the time needed for pretreatment antibiotics to diffuse out of cells.

Response: In addition to our above response, we would like to clarify that hysteresis is not entirely equivalent to drug interaction type. One may think of hysteresis as temporal drug interactions, where distinct inhibition effects are possible, depending on treatment direction. Most notably, our data reveal several directional effects of hysteresis. For instance, treatment with ceftazidime (CTZ) followed by gentamicin (GEN) results in negative hysteresis, whereas the reverse order leads to positive hysteresis—despite the drug combination exhibiting significant synergy (Figs. 2A, 2B). Furthermore, the drug

interaction profile cannot be inferred directly from the hysteresis effect, as examples in our data show: Any switch between CTZ and meropenem (MER) results in positive hysteresis, but their combination shows significant synergy (Figs. 2A, 2B).

These examples underscore that, while there is a relationship between negative hysteresis and synergy, the two phenomena are distinctly different. These points are now highlighted more clearly in the revised Results and the revised Discussion.

2. Experimental design and experimental method. The 'gold standard' method to measure the impact of antibiotics on bacteria viability is to measure how antibiotic exposure influences live bacteria cells counts. The authors use this approach to test for interactions between GEN and CAR in Figure 1. However, the experiments reported in this Figure are not replicated, making it impossible to assess the validity of these results.

Response: We fully agree with the reviewer regarding the use of CFU/mL as gold standard and would like to emphasize that we have used this approach throughout the manuscript, where samples sizes allowed for it. The initial set of experiments was designed to explore the directionality of effects and the influence of antibiotic concentration. We repeated the experiments and now included three independent experimental runs that show that concentrations below the minimal inhibitory concentrations of the sensitizing drug CAR were sufficient to induce negative hysteresis. We now explain these new replicated data sets in the revised Results section and present them in Fig1. B-D; we moved the original dose dependency test to the Supplement as new Supplementary Fig. S1A-C.

All experiments now always included at least 3 independent replicates per treatment, as explicitly indicated in the figure legends (see Figs. 1E, 1F, 1G, 2, 4, and 5) except for the screen (Fig 3), where we used 6-14 technical replicates for each strain. For the main conclusion of robustness, the different strains served as the independent replicates.

The authors then go on to test for hysteresis between pairs of antibiotics in clinical isolates. The authors based their measurements of interactions using OD curves. We appreciate that assays like this require high throughput experimental methods, but there are a number of important potential biases and limitations with using OD to measure responses to antibiotics. For example, exposure to B-lactam and fluoroquinolone antibiotics causes changes to bacterial cell morphology (ie cell filamentation) that alter the correlation between cell number and OD. Antibiotics also cause cell death, and OD does not allow for live and dead cells to be distinguished. Owing to these potential limitations, using OD to measure the impact of antibiotics on bacterial growth requires robust validations using methods that assess viable cell counts. We are not convinced by the validation provided by the authors, and we suggest that they should randomly test a fraction of the interactions in their screen instead of cherry picking 1 or 2 interactions, particularly if these are previously known examples of hysteresis.

Response: We thank the reviewer for this comment. We deliberately chose an OD-based approach using plate-readers to enable high-throughput screening across multiple *P. aeruginosa* strains and antibiotics. This scale of screening is not feasible with a CFU-based method, which we did use for all detailed analyses (Figs. 1B-G), 2A, 4B-D, 5A, C-D).

We agree that OD-based assays have potential confounding factors. Overall we would like to emphasize that we already acknowledged these in the extended methods of the original manuscript (see Supplement) and that the potential confounding factors such as filamentation or cell death are more likely to lead to an underestimation of negative hysteresis and an overestimation of positive hysteresis. For this reason, we focused our analysis and the discussion on negative hysteresis, while we remain cautious about conclusions regarding positive hysteresis. We now highlight these limitations more clearly in the revised Results section.

We now additionally provide validation for the validity of the OD measures by including a new data set of OD/CFU correlations across all our time-kill assays that compare CFU and OD600 at 6 hours after the main treatment in cuvettes and/or microtiter plates. This dataset reveals a linear relationship between OD and CFU, especially in cases where Gentamicin is the main treatment. We now refer to these in the revised Results and added a new supplementary figure (Supplementary Fig. S5)

3. The use of clinical isolates to test drug interactions is a positive aspect of this study, particularly given that drug interactions and collateral sensitivity are often poorly conserved across strains. However, only a subset of clinical isolates displayed 'robust' negative hysteresis, with only two COPD populations, many switches were neutral (lines 197-198). The manuscript would benefit from a clearer framework for why some isolates or resistance phenotypes fail to show the phenomenon

Response: We agree that only two pathogen populations from patients show statistically significant negative hysteresis, while several others exhibit a trend toward this effect. In the revised manuscript, we now acknowledge that significant negative hysteresis was only identified in two of the mixed-strain clinical populations. At the same time, we now also emphasize that the mPact panel includes 13 strains with clinical origin and specify these strains both in the revised Results section and in the legend to Fig. 3. Almost all of the clinical mPact strains show significant cases of negative hysteresis upon switches from beta-lactam antibiotics to aminoglycosides. Therefore, we can conclude that negative hysteresis is inducible in *P. aeruginosa* strains with clinical origins. This aspect is now highlighted in the Results section.

4. The authors identify the well-known Cpx envelope stress pathway as a key regulatory mechanism. Combining targeted gene deletions, overexpression constructs, and transcriptomic profiling provides a good methodological basis for mechanistic insights into how envelope stress leads to antibiotic sensitization. The authors provide convincing evidence that the cpx pathway is involved in antibiotic sensitization – however, the data for cpxS mutants (e.g., T163P, G188S) at times yield unexpected

outcomes (lines 235 237). Some mutants do not replicate the wild-type phenotype or lead to positive hysteresis. Clarification is needed on how these differences influence the interpretation that CpxS is a key driver of negative hysteresis. Additional biochemical or phenotypic assays (e.g., membrane permeabilisation assays) might help to confirm how each mutation modifies envelope stress responses.

Response: Thank you for this thoughtful comment. The involvement of CpxS in negative hysteresis requires that the negative-hysteresis phenotype observed in the wild-type strain is abolished in *cpxS* mutants. This is precisely what we observe: all tested *cpxS* SNP and indel mutants consistently lose the negative-hysteresis phenotype, while the PA14 wild-type strain assessed in parallel retains it (see Figs. 4B, 4C, and 4D). Interestingly, a loss-of-function *cpxS* deletion mutant still exhibits negative hysteresis (Figs. 4C, 4D), whereas *cpxS* overexpression abolishes the phenotype (Fig. 4D). Taken together, these results strongly support the idea that the identified and tested *cpxS* SNP and indel variants all represent gain-of-function mutations that eliminate negative hysteresis. The level of the abrogation of negative hysteresis then varies between the two *cpxS* SNP mutations (T163P, G188S) and the indel variant (DES81-83G), which is not unusual, especially considering that the SNP and indel mutations are localized in different parts of the protein (Fig. 4A).

Therefore, we politely disagree that our data yield unexpected outcomes. On the contrary, the data consistently demonstrate a role for *cpxS* in modulating negative hysteresis and support the conclusion that the identified SNP and indel mutations are gain-of-function changes that disrupt the sensitization effect. In response to the reviewer's comment, we now address these aspects in the revised Results and Discussion.

5. A further limitation of this study is that the statistical methodology is not described very clearly, making it difficult to assess the methods used to analyze the data. For example, it is unclear if the authors treated different measures of the same culture at different time points as independent observations or not (they should not be considered independent) and the degree to which their results were based on comparisons between biological and technical replicates.

Response: We apologize for the lack of clarity in our original description of the statistical analyses. We fully agree with the reviewer that data dependencies must be appropriately accounted for in statistical evaluations. We did exactly that – either by specifying the relevant factor in a multifactorial analysis of the data or, alternatively, by first summarizing dependent data across time (e.g., via calculating area under the curves, AUCs) and then using these summary statistics in subsequent analyses. Moreover, all our experiments specifically include biological replication, except for the screen where we tested technical replicates for the individual strains, and in the subsequent exploration of robustness, these strains served as the independent biological replicates. We now provide information on statistical analyses in the revised Methods, the revised Supplementary Methods, and the legends to figures. The revised Results further now explicitly refer to the supplementary tables containing the results of the statistical analyses.

Please also note that the relevant underlying datasets for statistical analyses (SI – Datasets) and the results of our statistical tests and analyses, which also include information on replication (SI – Tables), are provided in supplementary files.

Overall assessment: Our overall assessment is that we are not convinced that hysteresis is a real phenomenon or if the authors have effectively just developed a new assay to measure drug-drug interactions (point 1). This a fundamental problem with the study. The authors are forthcoming about how drug pharmacokinetics may make it difficult to apply sequential antibiotic treatment in the context of infections, but the strong dose dependency (ie comparing results with 1mg/L vs. 2 mg/L GEN in Figure 1B) and transient nature of these interactions (ie Figure 1F) also make us skeptical about the potential applications of this work for treating bacterial infections.

Response: We respectfully disagree with the reviewer’s conclusion that hysteresis is not a real phenomenon and trust that the reviewer will arrive at our conclusion when all facts are duly considered. We have comprehensive evidence that negative hysteresis arises from antibiotic-induced cellular and physiological changes, which subsequently enhance the killing efficacy of a second antibiotic – as already outlined above in our response to the first comment of the referee.

In addition, clinical applicability of negative hysteresis was not the objective of this study, which rather aimed at enhancing our general understanding of the phenomenon, which is currently missing and which is needed as a basis for any future research on this topic. We fully agree with the reviewer that great care and responsibility accompany any modifications to clinical practice or suggestions thereof. Here hysteresis actually shows several key advantages to other explored treatment modifications (old antibiotics are well known, effects are conserved ...). We would like to point the reviewer to the work of Guggenbichler (doi: 10.1016/s0140-6736(88)90226-7, cited in our manuscript) which may assuage some of the well-intended concerns, although it does not abolish the need for a more comprehensive clinical validation. We agree that these assays (i.e. 30 min of pre-treatment, washing, and main-treatment) are not applicable in patients and may raise skepticism but we would like to emphasize that the assays provided here are specifically designed to enable an in-depth characterization of the hysteresis phenomenon by avoiding retention of the pre-treatment antibiotic and running into a combination therapy. Therefore, we also phrase our statements on clinical applicability in a very cautious way and rather emphasize that based on our findings of robustness, negative hysteresis represents a promising focus for optimizing treatment designs. These additional details have been addressed in the revised discussion.

Reviewer #3 (Remarks to the Author):

Reviewer #4 (Remarks to the Author):

I co-reviewed this manuscript with one of the reviewers who provided the listed reports. This is part of

the Nature Communications initiative to facilitate training in peer review and to provide appropriate recognition for Early Career Researchers who co-review manuscripts.

Additional changes and improvements:

- 1) We now included Samarpita Banerjee as co-author, because she performed some of the new experiments. We also included her in the description of author contributions.
- 2) In numerous places in the Introduction, Results, and Discussion, we further refined the manuscript to enhance clarity and the logic of our manuscript.

REVIEWER COMMENTS

Reviewer #1 (Remarks to the Author):

My prior review was largely supportive and requested more clarity about methods and some greater context from the literature, which the authors addressed well.

However Reviewer 2 raised important methodological concerns that drew my interest. The biggest one is that the methods didn't preclude carryover of pretreatment antibiotic in the periplasm/cytoplasm of cells, so you can't distinguish whether NH is a real, post-drug effect, or an effect of residual intracellular drug. Reviewer 2 also points to the transience of NH as consistent with drug carryover, as the effect could be disappearing once the residual drug diffuses out of cells.

The authors did a lot of additional work to address this, but not a direct test (I think you'd need mass spectrometry to estimate intracellular drug concentration). While not totally satisfying, it convinced me that NH is real even if its underlying biology remains unclear. Specific points about this follow:

a. The authors did additional experiments where they wash pretreated cells 2x before their assay for NH. However, I don't think you can wash away intracellular drug.

Our response: Many thanks for raising this point. If a compound is solidly bound to cell components, then repeated washes will not remove the compound. If the compound is not strongly bound, then it may diffuse out of the cell over time, and repeated washes are likely to remove a major part of this unbound, extracellular compound. The used beta-lactam antibiotic carbenicillin is known to covalently bind to peptidoglycan-binding proteins (PBPs), which are enzymes that catalyze the cross-linking of peptidoglycans in the cell wall. Thus, even after repeated washes, the covalently bound carbenicillin molecules will not be removed. However, we expect that the unbound carbenicillin molecules will be removed and thus, newly expressed PBPs will not be affected by carbenicillin. Nevertheless, the covalently bound carbenicillin does compromise PBP function across time and, thus, it might contribute to the weakening of the cell wall and enhance the effect of the subsequently administered aminoglycoside antibiotic, thereby leading to negative hysteresis. The possible importance of covalently bound beta-lactam antibiotics has now been emphasized more clearly in the revised discussion of our manuscript. In the revised methods, we now also clarify that the washing experiment served to remove unbound carbenicillin and assess the resulting consequences.

b. They also added experiments showing that other membrane stresses can cause NH: 1) mild heat shock and 2) CCCP (depletion of membrane potential). From this they conclude that "abrupt changes to the membrane fluidity can trigger hysteresis". This proves their principle, but it's not a direct test.

Our response: This is a valid point. We now extended our discussion and added: "These observations are consistent with the idea that hysteresis builds up through dynamic changes to the cell envelope,

which are likely induced by β -lactam drugs covalently bound to PBP and the resulting reduction in cross-linked peptidoglycan and/or other β -lactam-like stress.”

c. They comment: “If antibiotic retention were the sole explanation, we would expect enhanced killing regardless of the second antibiotic used”. It is unclear to me why all drug combinations should produce the same outcome -- wouldn't this depend on mechanism of action?

Our response: Many thanks for this comment. We agree with the reviewer. Since the above argument has only been part of our previous response letter, but it is not an argument in the manuscript text, no changes are required.

Instead, in the results section of our manuscript, we argue more carefully: “Removal of possible residual pre-treatment antibiotic by two consecutive washes with phosphate buffered saline did not abolish a significant hysteresis effect (Supplementary Fig. S2B, statistical results in Table S2), consistent with negligible carry-over of unbound CAR into the main treatment. Covalently bound CAR and/or any CAR-induced changes in cellular physiology are thus likely to be sufficient for hysteresis.”

Moreover, in the discussion of our manuscript, we further argue that the results of the washing experiment, which should have removed unbound beta-lactams, indicated that residual unbound carbenicillin “play a negligible role for the case of CAR-GEN hysteresis”.

d. Finally, the authors argue that increased efflux would expel residual drug and abolish NH. They found that a mexR mutation increased resistance but had no effect on NH, suggesting there NH wasn't caused by intracellular drug. I believe this because they confirmed that the mutation caused increased efflux via resistance. If increased efflux doesn't affect NH, than NH can't all be due to residual drug.

Our response: Many thanks for supporting our arguments on this topic.

e. The authors don't clearly address that the transience of the hysteresis phenotypes may also be a corollary of residual drug diffusion. Rather, they say that NH is not the same as drug interactions because they get different NH types depending on the order of the drug, but it does not mean they are not related.

Our response: Many thanks for this point. However, in the previous version of the manuscript, we did already point out that the two phenomena are related but not identical.

Please see our statement in the previous version of the abstract: “Negative hysteresis and the Cpx system are linked in several cases to the expression of synergistic drug interactions, thus enhancing efficacy of antibiotic combinations.”

Please also see our discussion of the relationship between negative hysteresis and synergistic drug interactions in the previous version of the discussion: “Our findings further suggest that negative hysteresis is related, although not identical to the molecular processes underlying drug interaction synergy, as proposed previously (19). There is overlap, albeit not complete, in the antibiotic pairs causing negative hysteresis and synergy (**Fig. 2**). As observed before, several synergistic drug pairs (e.g. CTZ-GEN), produce a mild positive hysteresis in addition to the expected strong negative hysteresis. The

sequential experiment thus provides evidence for single-sided synergy induction and inducible protective responses (**Figs. 2A, 2B**). While the interaction profile could be inferred from hysteresis measurements for the CTZ-GEN pair, another example in our data shows that this is not always so: switches between CTZ and meropenem (MER) consistently produce positive hysteresis, but their combination shows synergy (**Figs. 2A, 2B**). These examples underscore that, while there is a relationship between negative hysteresis and synergy, the two phenomena are distinct. For interactions with GEN, the phenomena are clearly related, as the studied, CpxS T163P mutation abrogates cellular sensitization (**Fig. 4B-4D**) and simultaneously causes a reduction of synergy in antibiotic interactions (**Figure 4E**). This connection between antibiotic cellular sensitization and synergistic drug interactions is consistent with earlier data from 1962 on the cellular underpinnings of β -lactam – aminoglycoside synergy (i.e., synergy between penicillin and streptomycin) in *E. coli* (20)."

Because the particular aspects of the relationship between negative hysteresis and drug synergism have thus already been addressed in detail, we did not add any further discussion on this topic.

Reviewer #2 (Remarks to the Author):

The authors have made substantial changes to the content and text of the manuscript following peer review. Most of our concerns have been addressed, and the authors have substantially improved this manuscript, apart from one technical concern listed below. Addressing this concern will not substantially alter the conclusions of this manuscript. The only bigger issue that we have with the paper is the link between this work and the authors' previous paper on hysteresis (<https://www.pnas.org/doi/epdf/10.1073/pnas.1810004115>). It is clear that this paper has advanced our understanding of hysteresis, but we found ourselves in a position where we were questioning whether the advances in understanding presented in this paper were sufficient to warrant publication in a journal like Nat Comms. We think that this is a debatable point, and the editor may wish to consider this issue if it is raised as an issue by other reviewers.

Our response: We politely disagree with the reviewers' view. Our current study does provide a major advance in our understanding of the novel phenomenon of negative hysteresis, because (i) it provides the first evidence of the widespread and robust expression of negative hysteresis across genomic backgrounds for a particular pathogen taxon, (ii) it provides the first evidence of the robust expression of negative hysteresis upon switches from a beta lactam drug to an aminoglycoside which is beyond the previously reported switch from carbenicillin to gentamicin, (iii) it provides evidence that negative hysteresis can be induced in resistant strains and pathogen populations isolated from patients, thereby highlighting novel, currently not considered treatment options, (iv) it provides novel information on the molecular basis of negative hysteresis that relies on beta-lactam-induced changes in membrane fluidity and increased intra-cellular concentration of the subsequently administered aminoglycoside drug, and (v) it provides new evidence for the central role of the Cpx cell envelope stress response system in mediating hysteresis based on our detailed functional genetic analysis of mutant strains previously not available (e.g., loss-of-function, gain-of-function, overexpression mutants, as well as mutants of related genes). Overall, our manuscript draws attention to a phenomenon that has only been described

recently, that is largely unexplored as to its occurrence, the underlying molecular basis, and its high potential to improve antibiotic therapy. We anticipate that this manuscript will be of particular interest to microbiologists and infectious disease specialists working in basic research (because of the novel phenomenon) and also in an applied field (because of the high potential to improve antibiotic therapy).

Outstanding issue:

The authors have carried out paired measures of OD and CFUs to test the validity of their approach (Figure S5). When gentamicin is used as the main antibiotic for treatment the OD values and CFUs are well correlated, validating the use of OD. However, the correlation between microtiter plate OD and CFUs is not convincing in any way when B-lactams are used as the antibiotic for main treatment. The authors test for a correlation between logOD and logCFU by linear regression, and this analysis is flawed. The underlying data do not show a bivariate normal distribution (ie normally distributed x and y values) violating a central assumption of linear regression. Violating this assumption tends to lead to inflated r^2 values (for example when linear regression is on data sets that have two 'clouds' of points) and we argue that their estimated (low) r^2 value of 0.27 is an overestimate of the true strength of association – cultures with OD of 0.05 and .5 were associated with similar viable cell densities. The authors should drop all of the data from plate reader based assays where B-lactams were used as the main treatment.

Our response: Many thanks for this important point. As a response, we now repeated our analysis using a non-parametric approach, the Spearman correlation analysis, which does not require normality of the data. As before, we performed the correlation analysis separately for (i) data with beta-lactam antibiotics, versus (ii) all other cases. All of these correlation analyses still reveal a significant positive association between CFU and the linked OD data, indicating that the OD data is indicative for cell numbers, even for the treatments involving beta-lactam antibiotics.

Nevertheless, we still share the reviewers' concerns that OD values could be biased by cell filamentation induced by the beta-lactam antibiotics, but also by killed bacterial cells that have not lysed yet. These effects could have introduced a bias in any indication of positive hysteresis. On the contrary, any OD-based indication of negative hysteresis should then be considered more trustworthy, because the inference of negative hysteresis is based on the observation of lower OD values and because cell filamentation and/or non-lysed cells would rather artificially increase such values. In response to the reviewers concerns and to clearly acknowledge these points, we now extended the description of results:

“Importantly, the potential confounding factors of OD, such as antibiotic induced cell filamentation or cell death without lysis, are more likely to lead to an underestimation of negative hysteresis (i.e., OD values after pre-treatment are higher than expected from the number of alive cells due to presence of dead cells or filamentation, incorrectly suggesting low cell killing) and an overestimation of positive hysteresis. Therefore, following this reasoning, we consider the OD-based hysteresis screen to provide a conservative indication for the occurrence of negative hysteresis and a potentially biased indication of positive hysteresis, especially when β -lactam antibiotics were used for the main treatment. Accordingly, we focused our following evaluation of the screen's results on the cases showing negative hysteresis.”

Reviewer #3 (Remarks to the Author):

Reviewer #4 (Remarks to the Author):

Additional changes:

- (1) Since submission of the previous version of our manuscript, a very interesting article on the function of the Cpx system in *Pseudomonas aeruginosa* was published. We now integrated this article's findings that are of relevance to our study, into the revised discussion of our manuscript.
- (2) We updated the contact details for one of the co-authors, Badri Dubey, who is now based at the Department of Biosciences and Bioengineering, Indian Institute of Technology Dharwad, Dharwad, India.
- (3) We further improved English writing style in a few places across the manuscript, to improve clarity of our statements, yet without changing the meaning of the sentences.

Dear editor, dear all,

Many thanks for the additional suggestions from the editorial office how to further improve our manuscript. Our changes are described in our responses to the Author Checklist. We also provide below a response to the only comment from one of the reviewers (reviewer #2).

Many thanks for the general support!

On behalf of all authors,

Roderich Roemhild, Hinrich Schulenburg

Response to the comment of reviewer #2

The reviewer's comment is given in gray and italics, while our response is given in blue.

REVIEWERS' COMMENTS

Reviewer #2 (Remarks to the Author):

The authors have responded to my technical concerns, but I remain unconvinced by their arguments regarding the novelty and importance of this work, as outlined below and quoting from their response to referees.

(i) it provides the first evidence of the widespread and robust expression of negative hysteresis across genomic backgrounds for a particular pathogen taxon,

Demonstrating negative hysteresis in clinical isolates is an advance with respect to the authors' previously published work. However, I consider this to be an incremental advance as it is effectively confirming a previously reported finding. I think that the authors case here would have been much stronger if they had been able to uncover why clinical isolates differ in their antibiotic responses.

(ii) it provides the first evidence of the robust expression of negative hysteresis upon switches from a beta lactam drug to an aminoglycoside which is beyond the previously reported switch from carbenicillin to gentamicin,

I view this as being quite a minor point. Given the similarities in structure and mechanisms of action of different antibiotics from the same class it would be surprising if this negative hysteresis between B-lactams and aminoglycosides was not widespread. It would have been much more novel if, for example, the authors had been able to demonstrate negative hysteresis across a wide range of combinations of antibiotic class.

(iii) it provides evidence that negative hysteresis can be induced in resistant strains and pathogen populations isolated from patients, thereby highlighting novel, currently not considered treatment options,

I think that demonstrating the applicability of negative hysteresis to clinical isolates is important, but the authors present very little data on this in the manuscript – the data in Figure 3 is from isolates collected from n=3 patients!

(iv) it provides novel information on the molecular basis of negative hysteresis that relies on beta-lactam-induced changes in membrane fluidity and increased intra-cellular concentration of the subsequently administered aminoglycoside drug, and (v) it provides new evidence for the central role of the Cpx cell envelope stress response system in mediating hysteresis based on our detailed functional genetic analysis of mutant strains previously not available (e.g., loss-of-function, gain-of-function, overexpression mutants, as well as mutants of related genes).

I agree with the authors that they have substantial mechanistic detail to their previously published case of negative hysteresis. The mechanistic work reported by the authors provides a sufficient level of analysis as supporting work, but it does not rise to the level that I would expect, for example, from a paper on AMR mechanisms. Providing supporting mechanistic detail into a previously published phenomenon does not, in my view, constitute the kind of novelty that I expect from papers in Nature Communications.

Our response: We very much appreciate the thorough assessment of our work by reviewer #2 and the early career researchers, who contributed to the review. It is such rigor that helps to improve the scientific record. At the same time, we still politely disagree with the reviewer's conclusion that our work represents an incremental advance of knowledge. We would like to clarify that our previous publication provided the discovery and a baseline description of negative hysteresis – without any details of its exact characteristics. Our current study clearly goes beyond the previously published work. And, importantly, it does so along several lines simultaneously, such as: the robust occurrence of negative hysteresis across genomic backgrounds including patient isolates and patient-related populations (cf. Fig. 3A, 3D); its robust occurrence for switches from diverse and clinically relevant beta-lactam antibiotics to the aminoglycoside gentamicin (Fig. 2A, 3A-D); its robust occurrence in isolates resistant to the sensitizing drug (Fig. 3B, 3C); the previously unknown relationship between hysteresis and drug interaction type (e.g., synergism, additivity, and antagonism; Fig. 2B, 4E); the involvement of the CpxS envelope stress response system and especially several distinct *cpxS* gain-of-function variants in mediating negative hysteresis (Fig. 4A-4D); and the cellular characteristics of the phenomenon (Fig. 5). Please note that our study included *P. aeruginosa* isolates from not only 3 patients (as implied in the response of reviewer #2), but from a total of 16 independent patients. 13 independent patient isolates are part of the mPact panel, for which results are presented in Fig. 3A. Fig. 3D shows hysteresis results for strain mixtures (not single isolates) as obtained from 3 additional independent patients.

Overall, it is the combination of the insights obtained across these different lines that provide a major advance in our understanding of this largely unexplored phenomenon of negative hysteresis.